# Ctf4 organizes sister replisomes and Pol α into a replication factory

Zuanning Yuan[1], Roxana Georgescu[2,3], Ruda de Luna Almeida Santos[1], Daniel Zhang[3], Lin Bai[1], Nina Y Yao[3], Gongpu Zhao[4], Michael E O'Donnell[2,3]*, Huilin Li[1]*

[1]Structural Biology Program, Van Andel Institute, Grand Rapids, United States; [2]Howard Hughes Medical Institute, Chevy Chase, United States; [3]DNA Replication Laboratory, The Rockefeller University, New York, United States; [4]David Van Andel Advanced Cryo-EM Suite, Van Andel Institute, Grand Rapids, United States

**Abstract** The current view is that eukaryotic replisomes are independent. Here we show that Ctf4 tightly dimerizes CMG helicase, with an extensive interface involving Psf2, Cdc45, and Sld5. Interestingly, Ctf4 binds only one Pol α-primase. Thus, Ctf4 may have evolved as a trimer to organize two helicases and one Pol α-primase into a replication factory. In the 2CMG–Ctf4₃–1Pol α-primase factory model, the two CMGs nearly face each other, placing the two lagging strands toward the center and two leading strands out the sides. The single Pol α-primase is centrally located and may prime both sister replisomes. The Ctf4-coupled-sister replisome model is consistent with cellular microscopy studies revealing two sister forks of an origin remain attached and are pushed forward from a protein platform. The replication factory model may facilitate parental nucleosome transfer during replication.
DOI: https://doi.org/10.7554/eLife.47405.001

*For correspondence:
odonnel@rockefeller.edu
(MEO'D);
Huilin.Li@vai.org (HL)

Competing interests: The authors declare that no competing interests exist.

## Introduction

Replication of cellular genomes requires numerous proteins that work together in a replisome (*O'Donnell et al., 2013*). Replication in eukaryotes utilizes CMG helicase (Cdc45−Mcm2-7−GINS) (*Ilves et al., 2010*; *Moyer et al., 2006*; *Costa et al., 2011*), leading and lagging strand DNA polymerases (Pol) ε and δ, Pol α-primase, PCNA clamps, the RFC clamp loader, and numerous accessory factors of less defined function (*Bell and Labib, 2016*; *Burgers and Kunkel, 2017*). Replication in eukaryotes is performed in localized foci in nuclei that contain 10–100 replication forks (*Falaschi, 2000*; *Kitamura et al., 2006*). Recent super resolution microscopy has resolved these foci into single replicon factories from bidirectional origins (*Chagin et al., 2016*; *Saner et al., 2013*). Nuclear foci in the budding yeast, *Saccharomyces cerevisae*, most often contain only single replicon factories having two replication forks, although some foci consist of more than two replication forks (*Saner et al., 2013*). The molecular structure of a single core replicon factory unit, consisting of two replication forks is unknown, but several studies demonstrate that sister replication forks remain together during S phase (*Conti et al., 2007*; *Falaschi, 2000*; *Ligasová et al., 2009*; *Natsume and Tanaka, 2010*), and that sister double-strand (ds) DNA products are extruded in loops directed away from a replication protein scaffold (*Gillespie and Blow, 2010*; *Saner et al., 2013*).

Ctf4 (Chromosome Transmission Fidelity 4) is a homo-trimer (Ctf4₃) that connects Pol α-primase and CMG helicase (*Gambus et al., 2009*; *Miles and Formosa, 1992*; *Simon et al., 2014*; *Tanaka et al., 2009*). Previous EM studies reveal that Ctf4₃−Pol α-primase resides on the opposite side of CMG from the leading strand DNA polymerase (Pol) ε in an individual replisome (*Sun et al., 2015*) (*Figure 1a*). Ctf4 also transiently binds Dna2, Dpb2, Tof2 and Chl1, and thus Ctf4 is proposed to be a dynamic hub (*Villa et al., 2016*; *Samora et al., 2016*). Dynamic interaction enables multiple

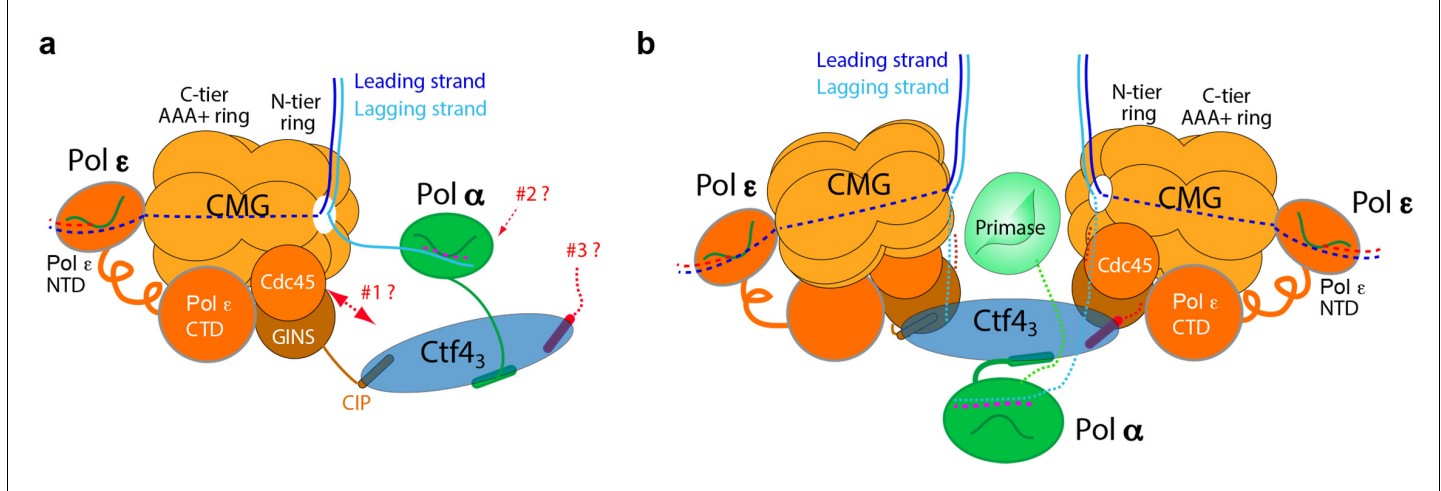

**Figure 1.** Comparison of individual replisomes and the coupled sister replisomes. (**a**) Individual replisome: Ctf4$_3$ connects CMG to Pol α-primase, and Pol ε binds the C-face of CMG. Three unanswered questions are represented by the three question marks: 1. What components in CMG, in addition to the CIP peptide of Sld5, interact with Ctf4$_3$ to maintain their stable association? 2. How does Pol α-primase interact with Ctf4$_3$ – on the top N-face or the bottom C-face of the Ctf4 disk? 3. Can a second Pol α-primase also occupy the third CIP-site of Ctf4$_3$? (**b**). Factory core model of two replisomes inferred from cryo-EM structures of the current report. Two CMGs bind tightly to two subunits of the Ctf4 trimer by an extensive interface formed with the Psf2 and Cdc45 subunits of CMG. One Pol α-primase occupies the third Ctf4 subunit. The single Pol α-primase connects to the C-face of Ctf4 near the split points of DNA entering the N-faces of the two CMGs. The two leading Pol ε complexes bind the C-face of CMG on the outside perimeter of the factory. See text for details.

DOI: https://doi.org/10.7554/eLife.47405.002

factors binding through time-sharing on Ctf4 and occurs through a conserved Ctf4-Interaction Peptide (CIP) motif (*Kilkenny et al., 2017*; *Samora et al., 2016*; *Simon et al., 2014*; *Villa et al., 2016*). The structures of the C-half of Ctf4 and its human orthologue AND1 reveal a disc-shaped constitutive trimer via the β-propeller domains on the N-terminal face (N-face) with the three C-terminal helical domains interacting with up to three CIP peptides on the C-face (*Kilkenny et al., 2017*; *Simon et al., 2014*). Ctf4 mutants that disrupt connection to CMG are deficient in transfer of parental nucleosomes to the lagging strand daughter duplex, with implications for a role in epigenetic inheritance during development (*Gan et al., 2018*). In human, AND1 (hCtf4) also binds CMG and is important for replication progression and DNA repair (*Kang et al., 2013*; *Abe et al., 2018*; *Williams and McIntosh, 2002*; *Yoshizawa-Sugata and Masai, 2009*).

Earlier structure studies of Ctf4 used subassemblies of CMG and Pol α-primase (*Simon et al., 2014*), and thus it has not been fully understood how Ctf4 interacts with the holoenzyme forms of CMG and Pol α-primase, and whether Ctf4 might bind two Pol α-primase as proposed (*Figure 1a*). We sought to address these questions by a combination of biochemistry and cryo-EM and found that one Ctf4$_3$ binds CMG very tight, not dynamic, and can bind 1, 2, or 3 CMG holoenzymes without steric hindrance at a 120° angle to one another. We also determine a structure of Ctf4$_3$ bound to Pol α-primase and observe Ctf4$_3$ binds only one copy of Pol α-primase. We have reconstituted a 2CMG–2Pol ε–1Ctf4$_3$–1Pol α-primase complex biochemically and can visualize a 2CMG–1Ctf4$_3$–1Pol α-primase complex by EM. Moreover, the CMGs retain helicase activity while multimerized by Ctf4$_3$. These findings led us to propose that sister-replisomes are coupled by Ctf4$_3$ in a 'replication factory' of 2CMG–Pol ε–1Ctf4$_3$–1Pol α-primase (*Figure 1b*). We further address in the Discussion how various CIP factors may bind the replication factory, how Pol α-primase may be utilized for lagging strand priming of sister replication forks, and implications of a factory for parental nucleosome transfer to daughter duplexes. We note that these in vitro findings require cellular validation, which will be pursued in a separate study.

# Results

## CMG–Ctf4 form a stable complex

To explore how $Ctf4_3$ interacts with replisome factors we performed glycerol gradient sedimentation of protein mixtures (*Figure 2—figure supplement 1*). This method originally revealed that Pol ε binds CMG, forming a CMG–Pol ε complex that sediments faster than either component alone (compare panels c and h with panel d) (*Langston et al., 2014*). CMG binding to $Ctf4_3$ was also readily apparent (compare panels c and g with panel e). It was initially surprising that the CMG–$Ctf4_3$ complex migrated heavier than CMG–Pol ε, even though $Ctf4_3$ is not quite as large as Pol ε, because studies in the human system indicated there was only room for one CMG on Ctf4 (*Kang et al., 2013*), consistent with an earlier proposal (*Simon et al., 2014*) (*Figure 2—figure supplement 1*, compare panels d and e).

To study the CMG–$Ctf4_3$ complex further we mixed $Ctf4_3$ and CMG and applied it to a MonoQ ion exchange column; a complex of CMG–$Ctf4_3$ eluted at > 400 mM NaCl (*Figure 2a*). This result indicated CMG−$Ctf4_3$ is a stable complex, and is not loose, consistent wth an apparent tighter interaction of GINS complex to $Ctf4_3$ compared to the CIP peptide of Sld5 (*Simon et al., 2014*). The MonoQ isolated CMG–$Ctf4_3$ complex was also stable to size-exclusion chromatography (SEC) (*Figure 2b*). We conclude that CMG is tightly bound and highly stable on the $Ctf4_3$ hub. Density scans of fractions within the SEC elution profile indicated a heterogeneous mixture of CMG–$Ctf4_3$

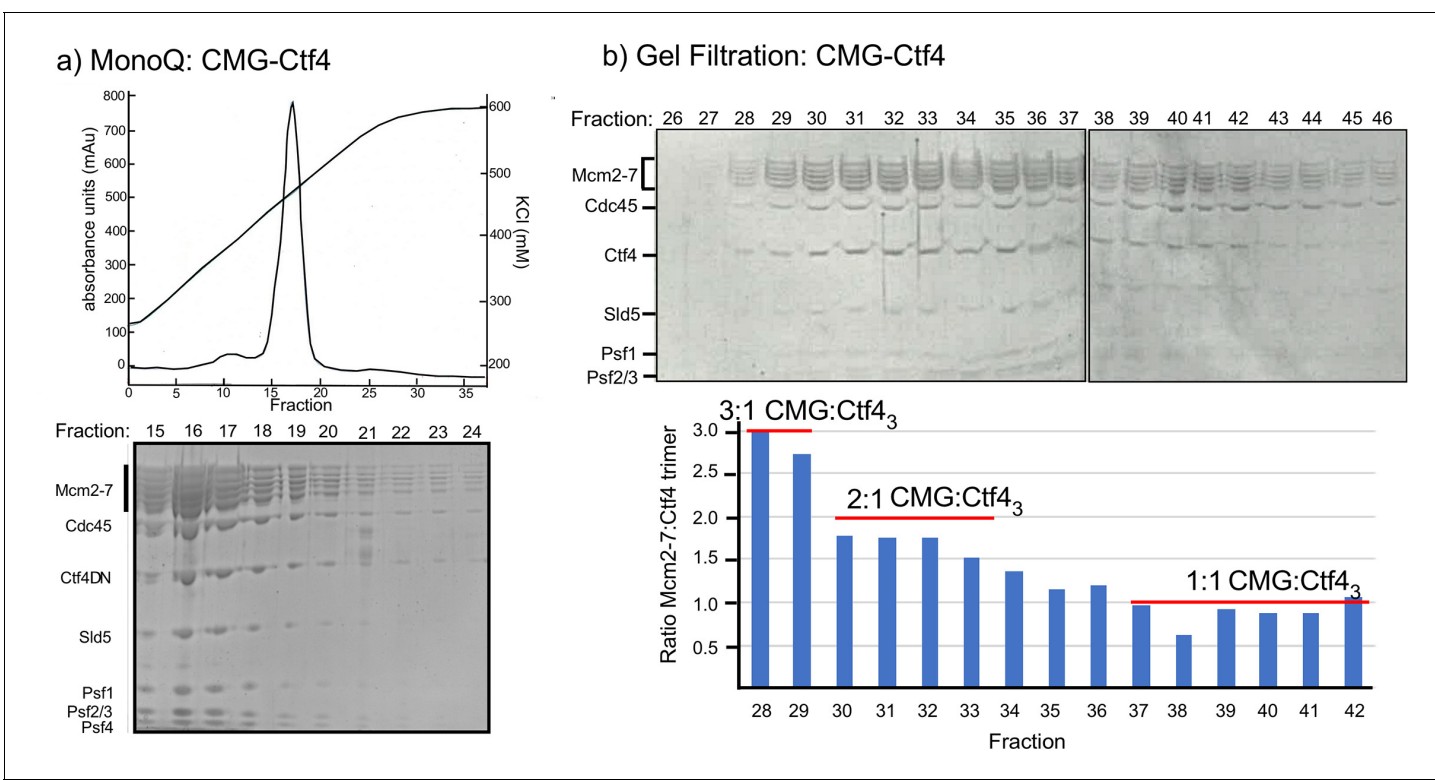

**Figure 2.** Multiple CMGs form a stable complex with the Ctf4 trimer. (a) A mixture of CMG and $Ctf4_3$ were applied to ion-exchange chromatography on a MonoQ column. The top panel shows the elution profile of the CMG−$Ctf4_3$ complex(s) which elute at approximately 450 mM NaCl. (b) The SDS-PAGE of gel filtration fractions from the MonoQ elution shows the CMG−Ctf4 complex(s) are stable during these chromatography steps. Ratios of CMG:$Ctf4_3$ as estimated by the gel density scans are indicated in the histogram. Glycerol gradient centrifugations also reveal a complex of CMG to $Ctf4_3$, as well as other complexes (*Figure 2—figure supplement 1*).

DOI: https://doi.org/10.7554/eLife.47405.003

The following figure supplement is available for figure 2:

**Figure supplement 1.** Reconstitution of replisome complexes.

DOI: https://doi.org/10.7554/eLife.47405.004

complexes, with CMG:Ctf4$_3$ ratios ranging from 3:1 to 1:1. We therefore examined different fractions by cryo-EM and found that indeed, more than one CMG can bind Ctf4$_3$, as described below.

## Structure of the 1CMG–Ctf4$_3$ complex

To investigate the structural basis underlying the strength of the CMG–Ctf4$_3$ complexes, we determined the structure of the 1CMG–Ctf4$_3$ complex by cryo-EM to 3.9 Å resolution. In agreement with previous structural studies of CMG (*Abid Ali et al., 2016*; *Yuan et al., 2016*), the C-tier AAA+ ring of Mcm2-7 is highly dynamic. By excluding this region in 3D refinement, we improved the 3D map of 1CMG–Ctf4$_3$ to 3.8 Å resolution (*Figure 3a–c*, *Table 1*, *Figure 3—figure supplements 1* and *2*).

Based on the density features, as well as previously reported separate structures of Ctf4$_3$ and CMG (*Simon et al., 2014*; *Yuan et al., 2016*), we built an atomic model (*Figures 3* and *4*, and *Figure 4—figure supplement 1*, *Video 1*). The overall architecture reveals an extensive interface between the helicase and Ctf4$_3$, amounting to 1706 Å$^2$ of buried area, larger than the stable contact between Cdc45 and Mcm2-7 (1182 Å$^2$) and between GINS and Mcm2-7 (1583 Å$^2$), explaining the stability of the CMG–Ctf4$_3$ complex. Interestingly, there is a wide gap between Ctf4 and the Sld5 subunit of CMG that contains the CIP peptide (*Figure 4a*).

There are three interfaces between CMG and Ctf4$_3$, involving three different subunits of CMG with one protomer of Ctf4$_3$ (*Figure 4c,d,e*; *Figure 4—figure supplement 2*). The two major interfaces are between the β-propeller region of Ctf4 and both the Cdc45 subunit of CMG (*Figure 4c*) and the Psf2 subunit in the GINS complex of CMG (*Figure 4d*); these interfaces were previously uncharacterized. The interactions between Cdc45 and the Ctf4 propeller are largely electrostatic, involving several salt bridges and hydrogen bonds. Furthermore, a loop connecting strands β1 and β2 of Psf2, disordered in the CMG structure in the absence of Ctf4$_3$ (*Yuan et al., 2016*), inserts into two blades of the β-propeller and becomes ordered by forming multiple interactions (*Figure 4d*). The interface between Psf2 and Ctf4 involves H-bonds between Psf2 residues Arg34 and Lys36 with Ctf4 residues Asn850 and Tyr848, respectively, as well as hydrophobic interaction among Ctf4 residues Phe518, Leu770, and Pro771 and Psf2 residues Ile27, Phe28, and Pro29. As expected, the main Ctf4-contacting regions of Cdc45, Psf2 are well conserved across evolution (*Figure 4—figure supplement 3*). The previously-identified interaction between Ctf4 and the CIP peptide in the Sld5 subunit (*Simon et al., 2014*) of the GINS complex is actually a minor interaction site in which the N-terminal residues 3–15 of Sld5 form a short helix that bundles with the helical domain of Ctf4, primarily via a hydrophobic interface (*Figure 4e*). An intervening long peptide (aa 16–53) of Sld5 is disordered. This long flexible linker to the CIP peptide of Sld5 may explain why the CIP peptide of Sld5 is not required to observe CMG binding to Ctf4 in cells, predicting a second tethering point between CMG and Ctf4 (*Simon et al., 2014*). Therefore the Psf2 and Cdc45 extensive interfaces explain the stable association between Ctf4 and CMG.

## The 2CMG–Ctf4$_3$ and 3CMG–Ctf4$_3$ complexes

Cryo-EM 2D averages of two CMGs bound to Ctf4$_3$ show the CMGs are held rigidly, consistent with their stable binding to Ctf4$_3$ (*Figure 5a*). The 3D reconstruction shows each CMG interacts with only one protomer of Ctf4$_3$, and the CMG contact is limited to one side of the Ctf4$_3$ triangle related by 120˚ (*Figure 5b*). While earlier 2D studies of Ctf4 binding the GINS subassembly also observed a similar geometry, it has been thought that only one large CMG holoenzyme could bind Ctf4 (*Kang et al., 2013*; *Simon et al., 2014*). However the structure of 2CMG–Ctf4$_3$ shows that the large CMGs do not sterically obstruct one another. Therefore, one Ctf4$_3$ may organize up to three CMGs. Indeed, we observed 2D class averages that show two or three CMGs per Ctf4$_3$, related by the three-fold axis of Ctf4$_3$ (*Figure 5a*, *Figure 5—figure supplements 1* and *2*). We determined the cryo-EM 3D maps of the 2CMG–Ctf4$_3$ complex at 5.8 Å resolution (*Figure 5b*, *Figure 5—figure supplement 1*, *Video 2*) and the 3CMG–Ctf4$_3$ complex at 7.0 Å (*Figure 5—figure supplement 2*), respectively. The interactions between individual CMG helicases and their respective partner Ctf4 protomers in both 2CMG–Ctf4$_3$ and 3CMG–Ctf4$_3$ complexes are virtually identical, the same as in the 1CMG–Ctf4$_3$ complex described above, consistent with no steric clash between the CMGs.

Considering that the 2CMG–Ctf4$_3$ complex may be more physiologically relevant, as factories consisting of two forks straddling one origin are observed in vivo (*Saner et al., 2013*), and a 2CMG

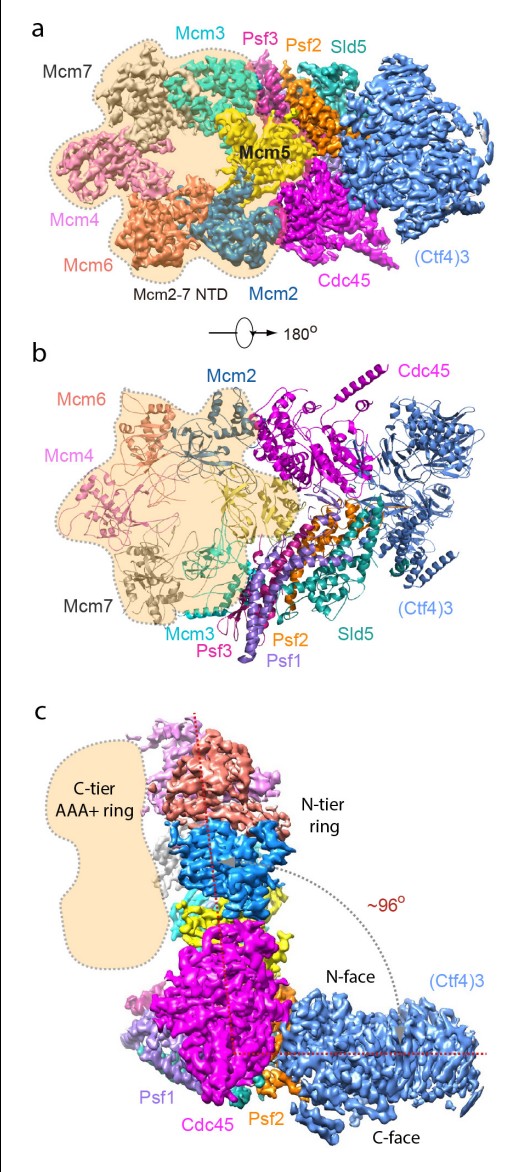

**Figure 3.** Cryo-EM structure of 1CMG–Ctf4₃. Cryo-EM density map of the 1CMG–Ctf4₃ complex in: (**a**) top view looking down the N-tier view of Mcm2-7. (**b**) Cartoon representation of the atomic model of 1CMG–Ctf4₃ in the C-tier AAA+ ring view. (**c**) Side view. Each subunit is colored differently. The Mcm2-7 AAA+ motor ring is flexible, and thus removed for higher resolution, but its position is shaded in beige. The resolution of this complex was facilitated by focused 3D refinement omitting the AAA+ C-tier of the Mcms in CMG (**Figure 3—figure supplements 1** and **2**).

DOI: https://doi.org/10.7554/eLife.47405.005

The following figure supplements are available for figure 3:

**Figure supplement 1.** Image processing and 3D reconstruction scheme.

DOI: https://doi.org/10.7554/eLife.47405.006

*Figure 3 continued on next page*

occupancy of Ctf4₃ leaves one protomer of Ctf4₃ for other CIP factors such as Pol α-primase, we continue this report in the context of the 2CMG–Ctf4₃ complex.

The key insight from the 2CMG–Ctf4₃ structure is that the two CMGs are held on the same side – the top side, or the N-face of the disc-shaped Ctf4₃ when the structure is viewed from the side of the Ctf4₃ disk (**Figure 5b**). Hence, the two N-tier rings of the Mcm2-7 hexamers – where the dsDNAs approach for dsDNA unwinding (**Georgescu et al., 2017**; **Douglas et al., 2018**) – approximately face one another at an angle of 120° and the C-tier motor rings of the two CMGs, where the leading strand Pol ε's bind (**Sun et al., 2015**), face outwards and away from each other.

## A 1:1 complex of Ctf4₃ and Pol α-primase

Earlier studies observed only one Pol α-primase bound to Ctf4₃ (**Simon et al., 2014**) (see their Figure 4e and Extended data Figures 8 and 9). We wished to understand the basis of this stoichiometry but the interaction of Ctf4₃ to Pol α-primase was too loose to isolate a complex for cryo-EM analysis, consistent with the dynamic hub model of Ctf4₃ (**Simon et al., 2014**; **Villa et al., 2016**). Thus we directly mixed Pol α-primase and Ctf4₃ at 1:1 and 3:1 molar ratios followed by cryo-EM analysis to study how Pol α-primase binds Ctf4₃ in the absence of CMG.

We first examined Pol α-primase alone and found Pol α-primase was flexible when frozen in vitreous ice, and did not generate well-defined 2D class averages. But under negative-stain EM conditions, Pol α-primase was sufficiently stabilized on carbon film to yield a bi-lobed shape with the two lobes ~ 120 Å apart (**Figure 6a**). This architecture is essentially the same as a previous negative-stain EM study, in which one lobe is assigned to the catalytic NTD of Pol1 (Pol1-NTD) and the other lobe to the CTD of Pol1, plus the B-subunit and the L- and S-subunits of the primase (**Núñez-Ramírez et al., 2011**). Cryo-EM of Ctf4₃ alone produced 2D class averages that are consistent with the crystal structure (**Figure 6b**). Cryo-EM 2D class averages of a 1:1 molar ratio mixture of Ctf4₃ and Pol α-primase yielded a structure comprised of Ctf4₃ bound to one catalytic Pol1-NTD of Pol α-primase (**Figure 6c**, **Figure 6—figure supplement 1**, **Table 2**). Increasing the Pol α:Ctf4₃ ratio to 3:1 did not change the 1:1 binding with Ctf4₃ (**Figure 6—figure supplement 2**). The presence of the Pol lobe, but absence of the primase lobe in the 2D class averages is consistent with previous

*Figure 3 continued*

**Figure supplement 2.** Image processing and resolution estimation of the 3D map of the 1CMG–Ctf4$_3$ complex.

DOI: https://doi.org/10.7554/eLife.47405.007

studies showing a high degree of flexibility between the primase and polymerase lobes (*Baranovskiy and Tahirov, 2017*; *Núñez-Ramírez et al., 2011*; *Perera et al., 2013*). The 2D averages reveal two contacts between Ctf4$_3$ and Pol1-NTD, but only one of the two contacts is visible in the 3D map (*Figure 6c–d*). These interactions must be weak and flexible, with one contact becoming averaged out in 3D reconstructions, and accounting for the low 12 Å resolution of the 3D map.

The crystal structures of Ctf4$_3$ and Pol1-NTD complexed with a primed DNA-dNTP fit well as two separate rigid-bodies in the upper and lower densities of the Ctf4$_3$–apo Pol1-NTD 3D map, respectively. The docking suggests that the primer-template duplex exits the Pol1-NTD from the bottom

**Table 1.** Cryo-EM 3D reconstruction and refinement of the three Ctf4$_3$–CMG complexes.

| | Ctf4$_3$–CMG$_1$ (EMD-20471) (PDB 6PTJ) | Ctf4$_3$–CMG$_2$ (EMD-20472) (PDB 6PTN) | Ctf4$_3$–CMG$_3$ (EMD-20473) (PDB 6PTO) |
|---|---|---|---|
| Data collection and processing | | | |
| Magnification | 130,000 | 130,000 | 130,000 |
| Voltage (kV) | 300 | 300 | 300 |
| Electron dose (e$^-$/Å$^2$) | 50 | 50 | 50 |
| Under-focus range (μm) | 1.5–2.5 | 1.5–2.5 | 1.5–2.5 |
| Pixel size (Å) | 1.074 | 1.074 | 1.074 |
| Symmetry imposed | C1 | C1 | C1 |
| Initial particle images (no.) | 759,267 | 759,267 | 759,267 |
| Final particle images (no.) | 200,491 | 53,853 | 53,117 |
| Map resolution (Å) | 3.8 | 5.8 | 7.0 |
| FSC threshold | 0.143 | 0.143 | 0.143 |
| Map resolution range (Å) | 3.5–5.0 | 5.0–8.0 | 5.0–8.0 |
| Refinement | | | |
| Initial model used (PDB code) | 3jc5, 4c8h | 3jc5, 4c8h | 3jc5, 4c8h |
| Map sharpening B factor (Å$^2$) | −146 | −135 | −143 |
| Model composition | | | |
| Non-hydrogen atoms | 34,366 | 90,831 | 131,141 |
| Protein and DNA residues | 41,92 | 11,221 | 15,710 |
| Ligands | 0 | 0 | 0 |
| R.m.s. deviations | | | |
| Bond lengths (Å) | 0.009 | | |
| Bond angles (°) | 1.46 | | |
| Validation | | | |
| MolProbity score | 2.05 | | |
| Clashscore | 10.96 | | |
| Poor rotamers (%) | 0.63 | | |
| Ramachandran plot | | | |
| Favored (%) | 91.65 | | |
| Allowed (%) | 8.16 | | |
| Disallowed (%) | 0.19 | | |

DOI: https://doi.org/10.7554/eLife.47405.008

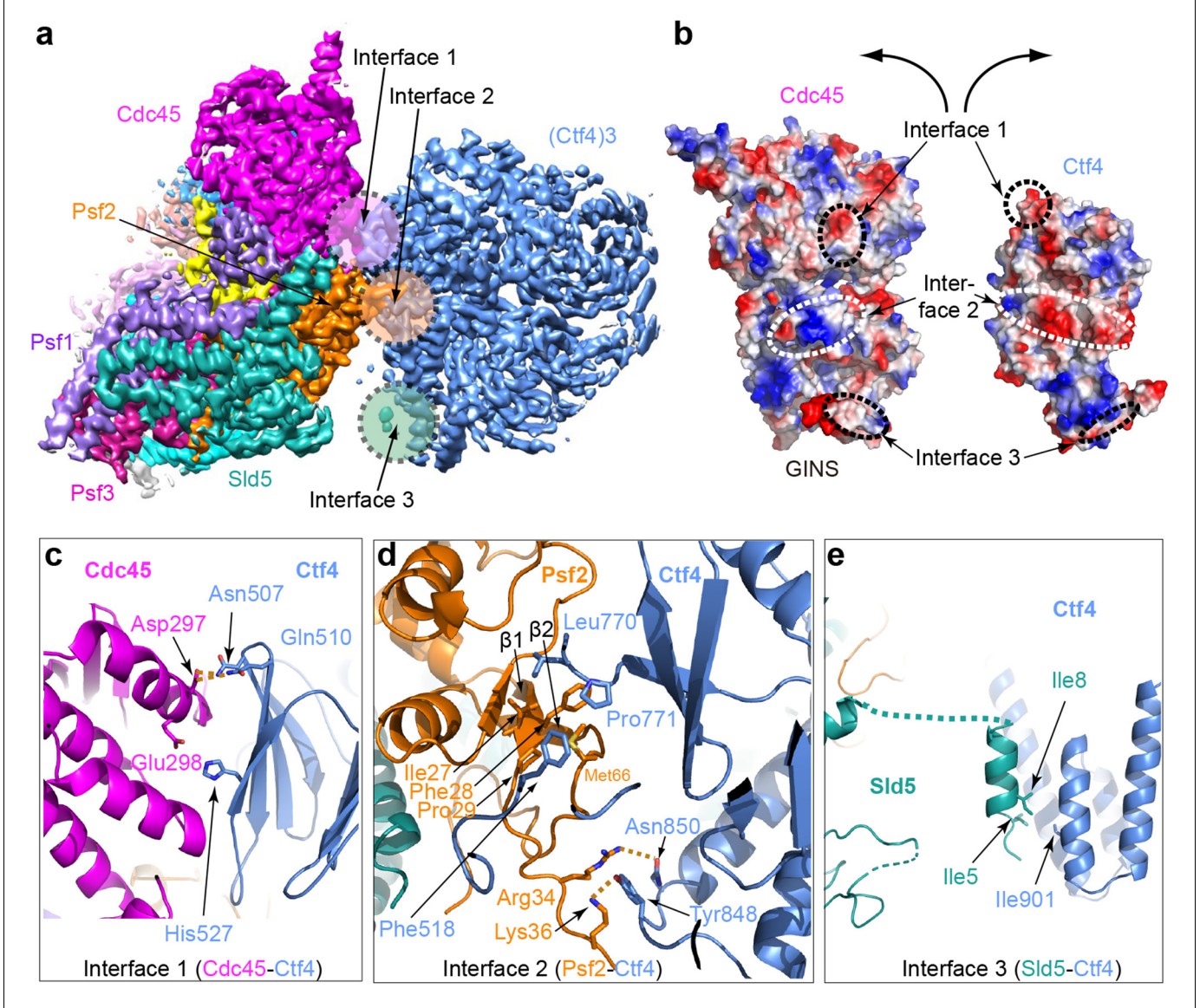

**Figure 4.** Molecular interface between Ctf4$_3$ and CMG. (a) 3D map showing the interacting Cdc45–GINS and Ctf4$_3$. Different subunits are in different colors as labeled, and the three interacting regions are labeled Interfaces 1 through 3. (b) An open-book view of the interface between Cdc45–GINS and Ctf4$_3$, shown in the electrostatic surface view. The three contacting regions between Cdc45−GINS and Ctf4 are marked by three pairs of dashed ellipses. (c–e) Interface one between Cdc45 and Ctf4 (c), interface two between Psf2 and Ctf4 (d), and interface three between Sld5 and Ctf4 (e), shown in cartoon view with several interacting residues shown in sticks. A close-up view of the interfaces is shown in *Figure 4—figure supplements 1* and *2*. Conservation in the interface region is shown in *Figure 4—figure supplement 3*.

DOI: https://doi.org/10.7554/eLife.47405.009

The following figure supplements are available for figure 4:

**Figure supplement 1.** Detailed densities at selected regions in the 3D map of Ctf4$_3$−CMG$_1$ complex.
DOI: https://doi.org/10.7554/eLife.47405.010

**Figure supplement 2.** Only one Ctf4 subunit engages with Cdc45 and GINS of one CMG helicase.
DOI: https://doi.org/10.7554/eLife.47405.011

**Figure supplement 3.** Sequence alignment of the Ctf4-contacting regions in CMG, Psf2 and Cdc45.
DOI: https://doi.org/10.7554/eLife.47405.012

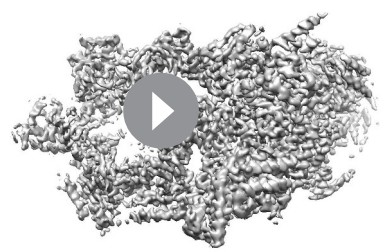

**Video 1.** A 360° rotation around a vertical axis of the 3D density map of 1CMG–Ctf4₃ at 3.8 Å resolution, followed by the subunit-segmented map, and the cartoon view of the atomic model. The map and model do not include the C-tier AAA+ motor ring of the Mcm2-7, which was excluded during 3D refinement.
DOI: https://doi.org/10.7554/eLife.47405.013

face (*Figure 6d*, *Figure 6—figure supplement 1*). The fact that use of a three-fold excess of Pol α-primase to Ctf4₃ did not provide more than one molecule of Pol α-primase bound to the Ctf4 trimer (*Figure 6—figure supplement 2*), is consistent with the previous observations (*Simon et al., 2014*). The Pol1-NTD occupies most of the bottom C-face of Ctf4₃ (*Figure 6d*, *Figure 6—figure supplement 1*), appearing to sterically occlude additional molecules of Pol α-primase and thus explaining the single Pol α-primase-to-Ctf4₃ stoichiometry. Notably, in a factory complex with two tightly bound CMGs, there would still remain a vacant CIP site in Ctf4₃ for binding of dynamic partner proteins, such as Pol α-primase and other CIP factors (see Discussion).

## Reconstitution and characterization of a 2CMG–Ctf4₃–1Pol α-primase complex

To investigate complex formation among CMG, Ctf4₃ and DNA polymerases, we analyzed by densitometry the sedimentation analyses of a mixture of CMG+Ctf4₃ with DNA Pol α-primase and Pol ε. This protein mixture produced a large super-complex that surpassed the size of CMG–Ctf4₃ and the Pol ε-CMG complexes (*Figure 2—figure supplement 1*). Interestingly, the bulk of excess Ctf4₃ is excluded from the large complex suggesting some type of cooperative assembly. Gel scans indicate a stoichiometry of two CMG–Pol ε, one Ctf4₃, one Pol α-primase (*Figure 7—figure supplement 1*), although we can't exclude a possible mixture of complexes. Upon mixing CMG+Ctf4₃+Pol α-primase we observed a 2CMG–1Ctf4₃–1Pol α-primase complex by negative stain EM (*Figure 7—figure supplement 2*).

We investigated cooperativity of CMG and Pol α-primase binding to Ctf4 by pull-down assays (*Figure 7a*). Pull-down assays using immobilized Ctf4₃ showed an 8-fold increase of Pol α-primase retained on Ctf4₃ when CMG was present, and conversely, more CMG bound to Ctf4₃ when Pol α-primase was present, suggesting cooperativity (*Figure 7a*). Cooperativity is consistent with densitometry of CMG–Pol ε–1Ctf4₃-–1Pol α-primase isolated in a glycerol gradient which excludes most of the Ctf4 trimer (*Figure 7—figure supplement 1*). Negative stain EM also shows a 2CMG–Ctf4₃–1Pol α-primase complex (*Figure 7—figure supplement 2*). The spontaneous and cooperative assembly of this complex in vitro suggests that the complex may also form in cells and possibly underlies the observations that sister replisomes are held together in cells (*Chagin et al., 2016*; *Saner et al., 2013*).

To determine if multimers of CMG bound to Ctf4 retain activity, and thus could operate on two forks at the same time, we performed helicase and replication assays comparing CMG with 2CMG + 1Ctf4₃ (preincubated to form 2CMG–Ctf4₃ complex as in *Figure 2*) (*Figure 7b,c*). Recent reports indicate that a 2 hr preincubation of CMG + DNA fork with 0.1 mM ATPγS or 0.5 mM ATPγS is necessary for CMG to bind forked DNA (*Burnham et al., 2019*; *Kose et al., 2019*). Therefore, we incubated CMG with forked DNA for 2 hr with ATPγS, which gave maximum binding (*Figure 7—figure supplement 3*). Helicase activity was initiated after the 120 min preincubation by adding 5 mM ATP. The results demonstrate that CMG helicase activity is slightly stimulated by Ctf4 compared to CMG alone (*Figure 7b*). Stimulation of CMG by Ctf4 was observed in an earlier study using human CMG and Ctf4 (*Kang et al., 2013*). We cannot distinguish whether Ctf4 stimulates intrinsic CMG helicase activity or enhances DNA binding (or both). Overall, we conclude CMG retains helicase activity while multimerized by Ctf4₃.

In *Figure 7c* we tested the effect of Ctf4 on replication activity with CMG and Pol α-primase. We first defined the time of preincubation needed for Pol α-primase to bind CMG +/- Ctf4 on the ³²P-primer forked DNA (*Figure 7—figure supplement 4*). The results indicate Pol α-primase locates the primed site within 1 min and thus we included a 5 min preincubation with Pol α-primase at the end of the 2 hr preincubation with ATPγS, CMG+primed fork DNA (+ /- Ctf4) prior to adding dNTPs and

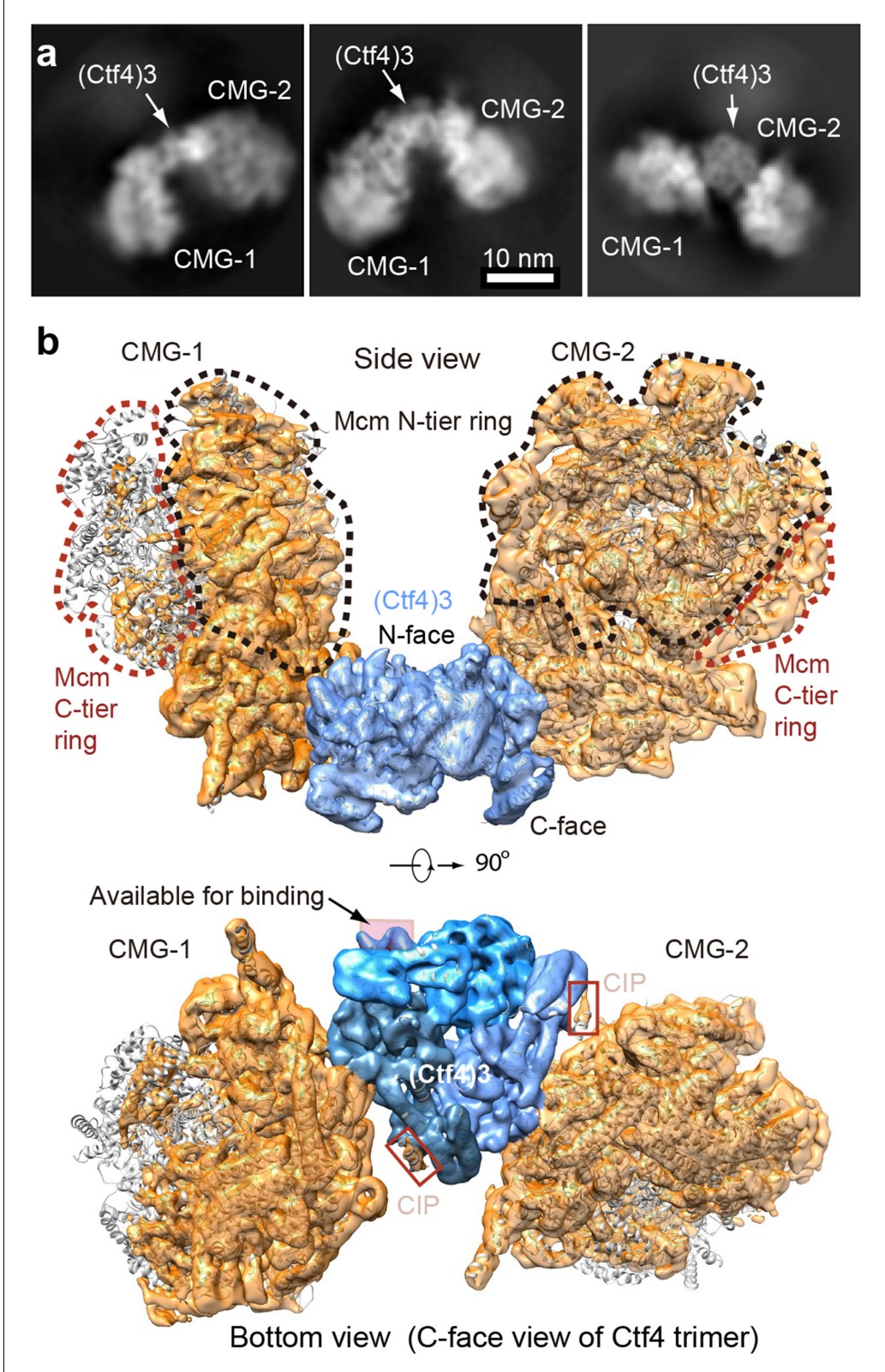

**Figure 5.** Cryo-EM structure of 2CMG–Ctf4$_3$. (a) Selected 2D class averages of cryo-EM images of 2CMG–Ctf4$_3$. (b) A side and a bottom view of the 3D map with docked atomic models of Ctf4$_3$ and CMG shown in cartoon view. The C-tier AAA+ ring of Mcm2-7 is partially flexible and the density is invisible at the surface rendering threshold used. See also *Figure 5—figure supplements 1* and *2* and *Video 2*.

DOI: https://doi.org/10.7554/eLife.47405.014

*Figure 5 continued on next page*

*Figure 5 continued*

The following figure supplements are available for figure 5:

**Figure supplement 1.** 3D map and resolution estimation of Ctf4$_3$ in complex with two CMG.

DOI: https://doi.org/10.7554/eLife.47405.015

**Figure supplement 2.** 3D maps and resolution estimation of Ctf4$_3$ in complex with three CMG.

DOI: https://doi.org/10.7554/eLife.47405.016

5 mM ATP (*Figure 7c*). Comparison of CMG+Pol α-primase with CMG+Ctf4$_3$+Pol α-primase showed that CMG retained full replication activity while multimerized by Ctf4$_3$, indicating that CMGs multimerized by Ctf4$_3$ are functional.

## Atomic model for a replication factory

In light of cell-based studies that observe that sister replisomes generated from a bidirectional origin are physically coupled such that the two sister duplexes extrude together away from a point source (*Chagin et al., 2016*; *Saner et al., 2013*; *Natsume and Tanaka, 2010*), and on the basis of our cryo-EM and biochemical analysis, we propose that the observed factory in cells is explained by one Ctf4$_3$ that coordinates two CMGs and one Pol α-primase to form a 2CMG–1Ctf4$_3$–1Pol α-primase core replisome factory. While we observed this core replisome factory in negative stain EM (*Figure 7—figure supplement 2*), we were unable to reconstruct the full 2CMG–Ctf4$_3$–Pol α-primase by cryo-EM analysis, and therefore we obtained an atomic model for this super-complex factory by superimposing the shared Ctf4$_3$ in the atomic model of 2CMG–Ctf4$_3$ with that of Pol α–Ctf4$_3$ (*Figure 8a–b*, *Figure 8—figure supplement 1*, *Video 3*). The model indicates a factory complex that contains two CMGs on the sides of the Ctf4 disk, and one Pol α-primase on the C-face of the Ctf4 disk. The protein complex consisting of 26 visually observed polypeptides in this > 2 MDa factory model give no steric clash among them. This model may explain why Ctf4 has evolved to assemble a trimer, not a dimer, in order to tightly coordinate two CMG helicases, leaving one Ctf4 protomer to bind transient CIP factors such as Pol α-primase. Such a structure, operating at a level above the individual replisome, may represent the functional unit of a cellular replicon core factory derived from one bidirectional origin as implicated by cellular and microscopy studies (*Chagin et al., 2016*; *Saner et al., 2013*). Our dimeric CMG replication factory model suggests possible coordination of the two forks that arise from an origin of replication. As the sister forks grow, the duplicated leading and lagging strands would form loops that push the nascent sister DNAs out from the factory surface in a scenario that resembles the proposed replication factory model based on cellular studies (summarized in *Figure 8—figure supplement 2*) (*Chagin et al., 2016*; *Saner et al., 2013*).

## Discussion

The factors that mediate the association of different replisomes and how a replication factory looks like have been unknown. The cryo-EM and biochemical studies presented here suggest a higher order architecture of the replication machinery beyond an individual replisome and propose that Ctf4 has evolved as a trimer to simultaneously organize two CMGs and one Pol α-primase by forming a 2CMG–Ctf4$_3$–1Pol α-primase factory. Our factory model conceptually advances on the previous view in which Ctf4$_3$ binds a single CMG of an individual replisome (*Figure 1a*) (*Simon et al., 2014*; *Villa et al., 2016*). However, we note that individual replisomes with 1CMG–Ctf4$_3$ and replisome factories with 2CMG–Ctf4$_3$ are not mutually exclusive and that until the factory is confirmed to operate inside the cell, the conclusions drawn here should be regarded as preliminary.

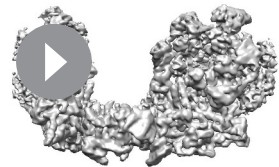

**Video 2.** A 360° rotation of the 3D density map of 2CMG–Ctf4$_3$, at 5.8 Å resolution, followed by density segmentation, and then the atomic model of 2CMG–Ctf4$_3$.

DOI: https://doi.org/10.7554/eLife.47405.017

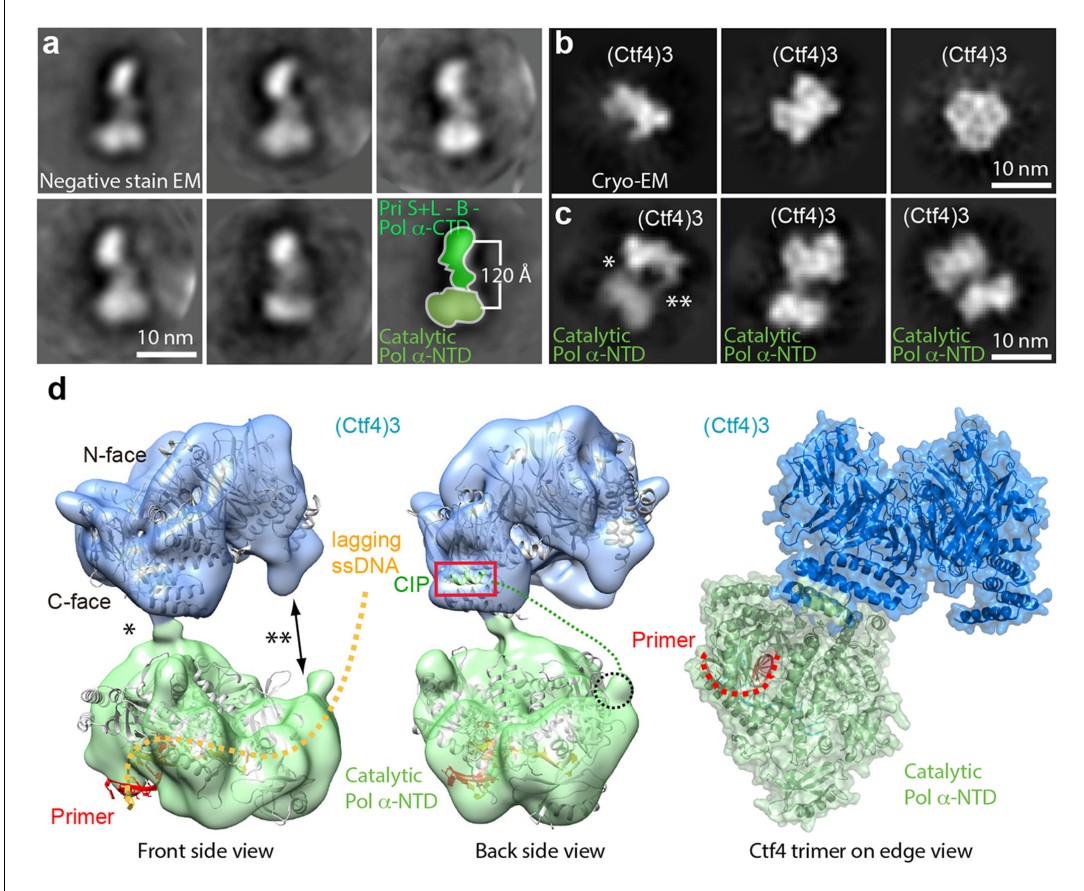

**Figure 6.** Cryo-EM map of Ctf4$_3$–Pol α-primase. (**a**) Selected 2D class averages of negatively stained images of Pol α-primase showing the enzyme in a similar view with a two-lobed architecture. (**b**) 2D class averages of cryo-EM images of Ctf4$_3$ in three distinct views. (**c**) Three selected 2D class averages of cryo-EM images of Ctf4$_3$−Pol α-primase. Note the primase lobe of Ctf4$_3$−Pol α-primase is not visible. (**d**) Left and middle panels: front and back views of the surface-rendered cryo-EM 3D map of Ctf4$_3$–Pol α-primase docked with the crystal structure of Ctf4 in light blue and crystal structure of the catalytic Pol α-NTD in light green. Right panel: atomic model viewed when the Ctf4 trimer is orientated horizontal and on edge. Rigid body docking is further presented in *Figure 6—figure supplement 1*. The asterisk (*) and double asterisk (**) in (**c, d**) mark the left and right contacts, respectively, between Ctf4$_3$ and Pol α-NTD. The right contact is not visible in the 3D map (**d**). Increasing the concentration of Pol α-primase did not give additional Pol α-primase bound to Ctf4 (*Figure 6—figure supplement 2*).

DOI: https://doi.org/10.7554/eLife.47405.018

The following figure supplements are available for figure 6:

**Figure supplement 1.** Cryo-EM of the Pol1–Ctf4$_3$ complex.

DOI: https://doi.org/10.7554/eLife.47405.019

**Figure supplement 2.** Only one Pol α-primase is observed to bind to the Ctf4 trimer using three fold excess Pol α-primase to Ctf4$_3$.

DOI: https://doi.org/10.7554/eLife.47405.020

## Mechanism of bidirectional replication by a replication factory

A factory complex containing a stable 2CMG–Ctf4$_3$ is consistent with cell-based studies and light microscopy of replicating DNA in *S. cerevisiae* cells (*Conti et al., 2007*; *Falaschi, 2000*; *Kitamura et al., 2006*; *Ligasová et al., 2009*; *Natsume and Tanaka, 2010*), where sister replication forks are shown by super resolution confocal microscopy with fluorescent markers on DNA to be physically associated within a twin fork replication factory at bidirectional origins and that the daughter duplexes are extruded together from a common protein platform (*Kitamura et al., 2006*; *Saner et al., 2013*). The model is also consistent with super resolution microscopy of mammalian replication nuclear foci revealing they are comprised of several single replication factories, each of which represents one bidirectional origin replicon (*Chagin et al., 2016*).

**Table 2.** Cryo-EM 3D reconstruction of the Ctf4₃–Pol α-primase complex.

| | Ctf4$_3$–Polα-primase (EMD-20744) |
|---|---|
| Data collection and processing | |
| Magnification | 120,000 |
| Voltage (kV) | 200 |
| Electron dose (e⁻/Å²) | 60 |
| Under-focus range (μm) | 1.5–2.5 |
| Pixel size (Å) | 1.21 |
| Symmetry imposed | C1 |
| Initial particle images (no.) | 237,688 |
| Final particle images (no.) | 48,414 |
| Map resolution (Å) | 12 |
| FSC threshold | 0.143 |
| Map resolution range (Å) | 10–15 |

DOI: https://doi.org/10.7554/eLife.47405.021

In the 2CMG factory model (*Figure 8*), the two helicases stand sideways above the Ctf4$_3$ disk, with their respective N-tier rings of the two Mcm2-7 hexamers facing approximately 120° to each other. Previous studies show that the leading strand enters the N-tier of CMG and proceeds through the central channel of CMG to engage the leading strand Pol ε located at the C-tier of CMG (*Georgescu et al., 2017*; *Goswami et al., 2018*). Therefore, at the core of the replication factory, the two parental duplexes enter their respective CMGs at the N-tier where each duplex is unwound by steric exclusion (*Fu et al., 2011*; *Georgescu et al., 2017*; *Langston et al., 2017a*; *Langston and O'Donnell, 2017b*; *Goswami et al., 2018*; *Kose et al., 2019*; *Eickhoff et al., 2019*). In the core replication factory model, two parental duplexes can easily be engaged at the N-tiers of each CMG due to their 120° orientation, and the lagging strand is deflected off the top of the N-tier ring after embrace of the parental duplex by the zinc fingers that encircle dsDNA (*Georgescu et al., 2017*; *Li and O'Donnell, 2018*; *O'Donnell and Li, 2018*; *Goswami et al., 2018*). Therefore, the unwound leading strands travel through the horizontal central channels of CMGs to exit the left and right sides of the replication factory at their respective CMG C-tier to which the leading Pol ε binds.

## Pol α-primase is flexibly associated in the replisome

Given the high degree of flexibility between the Pol and primase lobes of Pol α-primase, it seems likely that only one lobe or the other will be observed in the EM depending on which lobe is more 'fixed' in place. In an earlier study that contained CMG–Pol ε–Ctf4–Pol α-primase-forked DNA, a density from Pol α-primase was observed at the N-tier of the Mcm ring (*Sun et al., 2015*). Given that the p48/p58 primase subunits interact with Mcm3, Mcm4, and Mcm6 (*You et al., 2013*), the Pol α-primase density was most likely the primase lobe. Pol1 of Pol α-primase binds directly to Ctf4$_3$ (*Simon et al., 2014*), and in the present study the Ctf4–Pol α-primase structure is solved in the absence of CMG and DNA. Therefore, it is the Pol1 lobe instead of the primase lobe that is stabilized on Ctf4. Thus, the current study and the earlier study are compatible, but observe different lobes of Pol α-primase. At the time of the earlier work (*Sun et al., 2015*), the dimeric replisome images were observed but discarded, because we did not know how to interpret those images.

## Either one or more Pol α-primases may prime the two lagging strands of the coupled sister forks

The lagging strands displaced off the N-tier rings of their respective Mcm2-7 hexamers are near the center between the two CMGs. Such lagging strand positioning allows for and raises the possibility that the single Pol α-primase may prime both lagging strands. The catalytic Pol1 NTD of Pol α-primase binds to the bottom C-face of the Ctf4$_3$ disk (*Figure 6d*, *Figure 6—figure supplement 1*). The

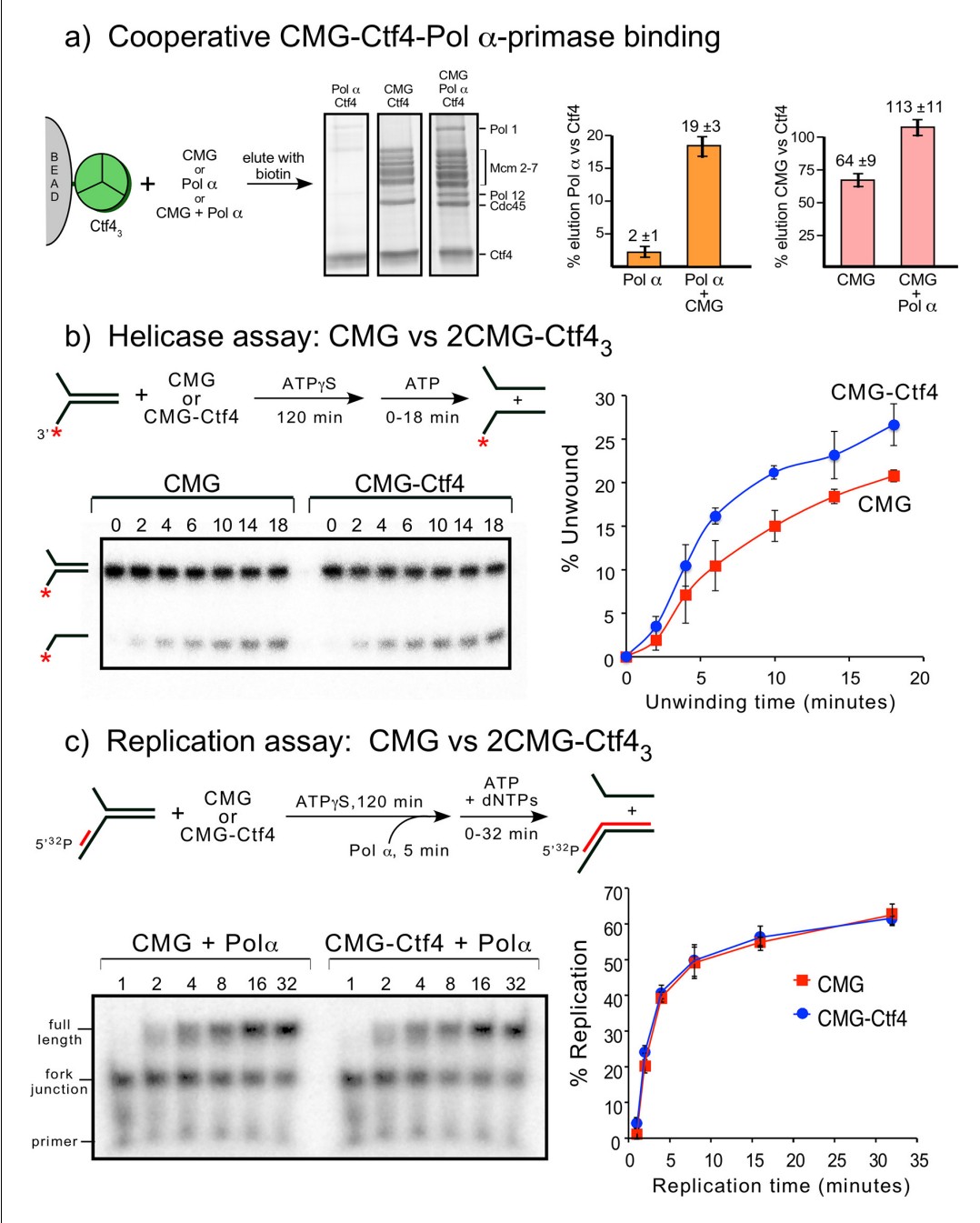

**Figure 7.** Cooperative assembly of the CMG−Ctf4₃−Pol α-primase complex, and the demonstration that CMGs multimerized by Ctf4₃ have helicase activity and support DNA replication. (**a**) Streptag-Ctf4 trimer on streptactin magnetic beads was added to either Pol α, CMG or a mixture of Pol α + CMG. Proteins were eluted with biotin and analyzed by SDS-PAGE (upper right). The assay was repeated in triplicate and gel scans were quantitated (below). The error bars show the standard deviation. The cooperativity is consistent with the stoichiometric assembly of 2(CMG–Pol ε)−1Ctf4₃−1Pol α-primsae while excluding most of the Ctf4₃ in a glycerol gradient analysis (**Figure 7—figure supplement 1**). The results are also consistent with negative stain EM of a mixture of CMG+Ctf4₃+Pol α-primase, showing the presence of a 2CMG−1Ctf4₃−1Pol α-primase complex (**Figure 7—figure supplement 2**). (**b, c**). Both CMGs in the 2CMG−Ctf4₃ factory are active. (**b**) Controls for testing CMG−DNA binding time in 0.1 mM ATPγS are in **Figure 7—figure supplement 3**. Native PAGE analysis of helicase assays upon preincubation of either CMG or CMG + Ctf4 for 2 hr with 0.1 mM ATPγS and a ³²P-forked DNA followed by 5 mM ATP to initiate unwinding. Timed aliquots were removed for analysis as indicated in the representative native PAGE gels. The plot represents results of triplicate assays. The mean value is indicated by the symbols and error bars show one standard deviation. (**c**) Preincubation experiments to determine the time for Pol α-primase to assemble onto the forked DNA are shown in **Figure 7—figure supplement 4**. Either CMG or CMG+Ctf4₃ were preincubated with ³²P-primed DNA fork and 0.1 mM ATPγS for 115 min, followed by addition of Pol α-primase and a further 5 min incubation before initiating replication/unwinding with 5 mM ATP and 0.1 mM each dNTP. Timed aliquots were removed

*Figure 7 continued on next page*

*Figure 7 continued*

for analysis as indicated in the representative native PAGE gels. DNAs having CMG bound enable Pol α-primase to extend the DNA to full length. Pol α-primase only extends to the forked junction on DNAs that lack CMG due to inability of Pol α-primase to perform strand displacement synthesis. The plot of full-length products represents results of triplicate assays. The mean value is indicated by the symbols and error bars show one standard deviation.

DOI: https://doi.org/10.7554/eLife.47405.022

The following figure supplements are available for figure 7:

**Figure supplement 1.** Densitometry analysis of CMGE—Ctf4$_3$—Pol α-primase.

DOI: https://doi.org/10.7554/eLife.47405.023

**Figure supplement 2.** EM observations of a 2CMG—Ctf4$_3$—1-Pol α-primase complex.

DOI: https://doi.org/10.7554/eLife.47405.024

**Figure supplement 3.** Establishing preincubation conditions for CMG binding to DNA before adding ATP for unwinding in helicase assays.

DOI: https://doi.org/10.7554/eLife.47405.025

**Figure supplement 4.** Establishing preincubation time for Pol α-primase binding to $^{32}$P-primed DNA for replication with CMG+/-Ctf4.

DOI: https://doi.org/10.7554/eLife.47405.026

primase lobe is not visible in the 3D map of Ctf4$_3$–Pol α-primase of this report, but the Pol1-NTD lobe and primase lobe of the bi-lobed Pol α-primase are known to be separated by a distance of ~ 120 Å and connected via a flexible tether that provides a 70° range of motion between the primase and polymerase lobes, sufficiently long to contact both CMGs in the factory (*Figure 8*, and *Figure 8—figure supplement 1*) (*Baranovskiy and Tahirov, 2017*; *Núñez-Ramírez et al., 2011*; *Perera et al., 2013*). Because the primase functions upstream of the Pol1 subunit of Pol α, and binds the Mcms (*You et al., 2013*), it is likely that the primase lobe extends upwards passing the 45 Å-thick Ctf4$_3$ disc to reach past the N-face of Ctf4$_3$, placing the primase between the two N-tier rings of the helicases where the two lagging strands are first produced (*Figures 1b* and *8a,b*). Therefore, the primase subunits can come in contact with and thereby prime both lagging strands.

However, we do not expect that only one Pol α-primase molecule functions for both lagging strands in a replication factory for several reasons. First, Pol α-primase is fully competent to prime the lagging strand in the absence of Ctf4 in vitro using pure proteins, and is dependent on CMG not Ctf4 for function (*Georgescu et al., 2015b*; *Yeeles et al., 2015*; *Yeeles et al., 2017*), indicating that Ctf4 is not required for replication fork operations per se. Second, the dynamic binding of Pol α-primase to Ctf4 in the dynamic Ctf4 hub view (*Villa et al., 2016*), especially given the weak binding of the Pol1 CIP peptide, would only enable a single Pol α-primase to stay bound to Ctf4 for a few seconds or less. Therefore, even individual replisomes containing Ctf4 would utilize numerous Pol α-primase binding events over the time needed to repeatedly prime one lagging strand during replisome progression.

## Multiple CIP factors may still access Ctf4 in the replisome factory

The currently identified CIP factors include Pol α-primase, CMG, Chl1, Dpb2, Tof2, and Dna2; the Pol α-primase, CMG, Dna2, and Chl1 bind the same consensus CIP site of Ctf4, whereas Tof2 and Dpb2 bind a distinct site on Ctf4 (*Samora et al., 2016*; *Villa et al., 2016*). The ability of Ctf4$_3$ to bind several different factors is proposed to depend on time-sharing due to weak CIP peptide binding with rapid on/off rates to Ctf4, similar to PCNA binding factors via a conserved PIP (PCNA Interaction Peptide) motif, reviewed in *Georgescu et al. (2015a)*. Given the tight interaction of 2 CMGs to Ctf4$_3$, we envision three different pathways for CIP proteins to bind Ctf4$_3$ in a replication factory. The first, and simplest, is that the transient binding of Pol α-primase will often vacate one Ctf4 subunit, making it available for other CIP proteins. Indeed, given Pol α-primase still functions in vitro without Ctf4 (*Georgescu et al., 2015b*; *Yeeles et al., 2015*; *Yeeles et al., 2017*), this particular CIP site may often be available for other CIP factors to bind. Second, the Chl1 helicase and Dna2 nuclease are required under particular cellular conditions, during which the replisome may change composition, and this has precedence in the rapid and dynamic rearrangements of proteins in the *E. coli* replisome (*Indiani et al., 2009*; *Lewis et al., 2017*). Third, the structures of this report show that the CIP peptide of CMG (in Sld5) is located across a wide gap between CMG and Ctf4, and the CIP sequence only contacts Ctf4 at the end of a long flexible linker in Sld5. The K$_d$ of the Sld5 CIP peptide to Ctf4 is only 5 μM and can even be deleted without preventing CMG–Ctf4 interaction in living

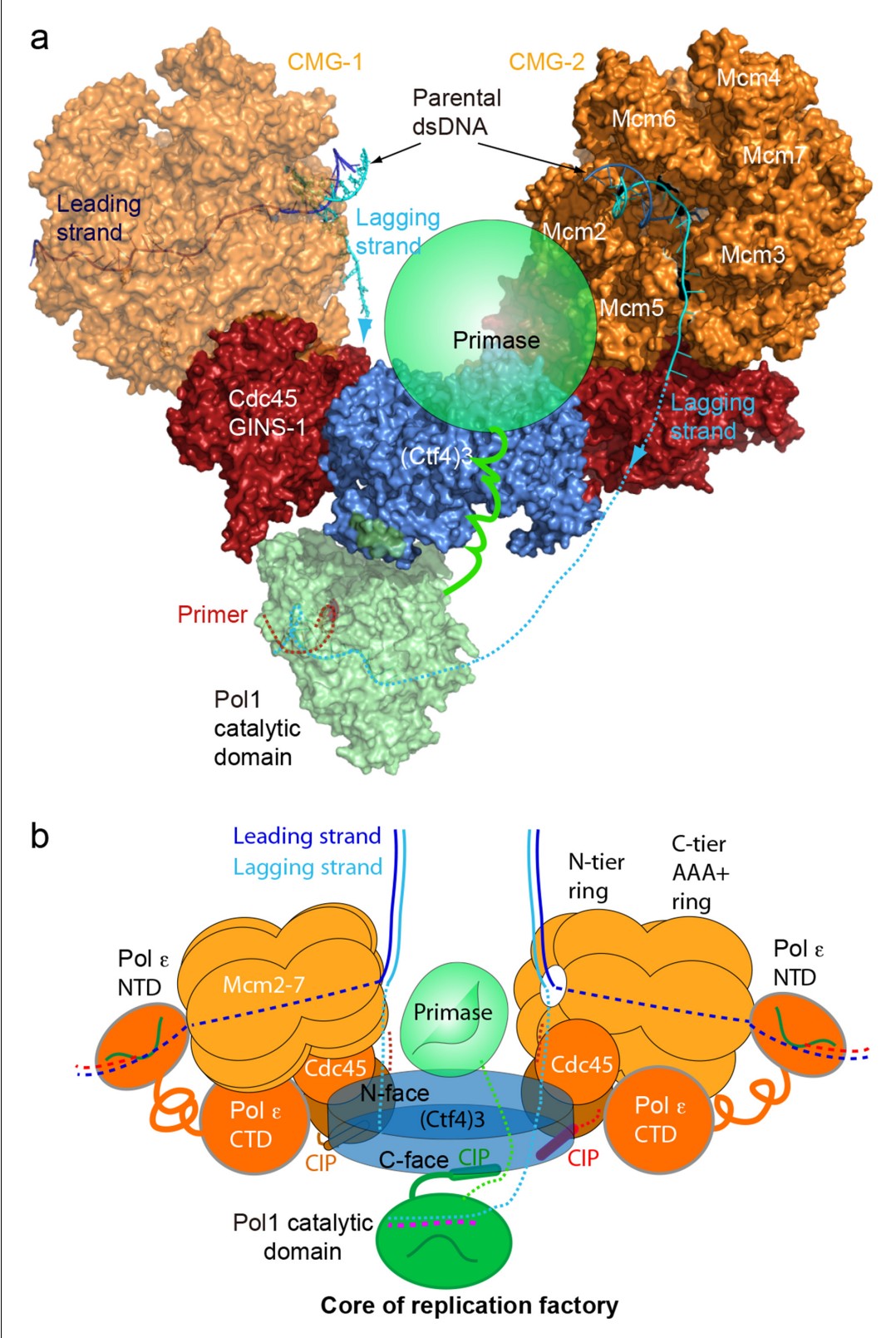

**Figure 8.** A model for the coupled sister replisomes. (**a**) A composite atomic model of one Pol α-primase and two CMG helicases organized in a core factory with a Ctf4 trimer. The model is derived by aligning Ctf4₃ shared between the Ctf4₃–CMG dimer model and the model of Ctf4₃–Pol α NTD. The DNA structure is based on the structure of CMG–forked DNA (PDB 5U8S), but the lagging strand outside the CMG channel is modeled. The possible location of the primase module of Pol α-primase is indicated by a green ellipse. (**b**) A sketch illustrating the leading strand Pol ε at the C-tier face of the

*Figure 8 continued on next page*

*Figure 8 continued*

CMG helicase and the primase reaches atop the N-face of Ctf4$_3$, potentially capable of priming both lagging strands. See text for details. See also
*Figure 8—figure supplements 1* and *2*, and *Video 3*. Figure Supplements and their legends.

DOI: https://doi.org/10.7554/eLife.47405.027

The following figure supplements are available for figure 8:

**Figure supplement 1.** The Pol1 and primase lobes of Pol α-primase have a 70° range of motion.

DOI: https://doi.org/10.7554/eLife.47405.028

**Figure supplement 2.** Comparison of the proposed sister replisome core factory to conclusion of super-resolution imagine of marked DNA in cells.

DOI: https://doi.org/10.7554/eLife.47405.029

cells (*Simon et al., 2014*). Given the flexible loop that mediates the Sld5 CIP peptide in the CMG–Ctf4 complex, the Sld5 CIP peptide likely retains the rapid $k_{off}$ implied by the 5 μM $K_d$ and thus can be expected to vacate the Ctf4 CIP site frequently (i.e. milliseconds). While speculative, it is also possible the Sld5 CIP peptide within CMG may be regulated by other proteins, or that other CIP factors that have additional contacts to Ctf4 that can outcompete the weakly bound Sld5 CIP peptide.

## Independent replisomes and the twin CMG factory model are not mutually exclusive

The current report demonstrates that two (or three) CMG can bind Ctf4 tightly, that CMGs bound to the Ctf4 trimer retain activity, and that a complex of 2(CMG–Pol ε)–1Ctf4$_3$–1Pol α-primase spontaneously assembles in vitro. In vitro single molecule studies in *S. cerevisiae* extracts and *Xenopus* extracts demonstrate that individual replisomes can move apart in opposite directions from an origin and contain only one CMG apiece (*Duzdevich et al., 2015*; *Yardimci et al., 2010*). While these experiments reveal that replisomes can act individually, these experiments utilize DNA tethered at both ends and thus DNA looping needed in our factory model would not be observed. Thus if factories were present, only when they dissociate to form independent forks would replication forks on doubly tethered DNA be visualized. Alternatively, Xenopus egg extracts may be programmed to replicate a bit differently from normal cells. It is worth noting that neither the model of an individual replisome nor the model of a dimeric replisome factory have been proven to exist inside cells. Because the same proteins are present in both models, they may not be mutually exclusive and may both exist in cells. If true, the different models may even fulfill distict functions. Clearly, cellular studies are needed to untangle these scenarios.

## Implications of a replisome factory on nucleosome distribution.

During replication the epigenetic marks on nucleosomes are distributed nearly equally to the two daughter duplexes (*Gan et al., 2018*; *Petryk et al., 2018*; *Yu et al., 2018*). The general view is that the H3H4 tetramer, which carries the bulk of epigenetic marks, binds DNA tightly and is transferred to nascent DNA independent of H2AH2B dimers that are easily displaced and exchanged (*Alabert et al., 2017*). Two binding sites for H3H4 exist in the replisome: The N-terminal region of Mcm2 binds H3H4 (*Huang et al., 2015*; *Richet et al., 2015*), and the Dpb3/4 subunits of Pol ε bind H3H4 at the C-face of CMG (*Bellelli et al., 2018*; *Sun et al., 2015*). Recent studies have found specific roles of these H3H4 sites in transfer of parental histones to daughter DNAs during replication (*Gan et al., 2018*; *He et al., 2017*; *Petryk et al., 2018*; *Yu et al., 2018*; *Evrin et al., 2018*). Moreover, the N-terminal region of Pol1 (i.e. the DNA Pol

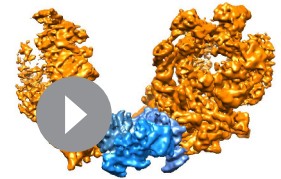

**Video 3.** Combination of the structures of 2CMG−Ctf4$_3$ with Ctf4$_3$–Pol α-primase to generate a pseudo-atomic model of the 2CMG−Ctf4$_3$–1Pol α-primase core replicon factory, by overlapping the shared Ctf4$_3$ density of the 2CMG−Ctf4$_3$ and Ctf4$_3$–Pol α-primase structures. The final model is rotated 360° around a vertical axis.

DOI: https://doi.org/10.7554/eLife.47405.030

of Pol α-primase) binds H2A/H2B, and mutations in either Pol1, Mcm2 or Ctf4 have a negative effect on transfer of parental histone marks to the lagging strand duplex (*Evrin et al., 2018*; *Gan et al., 2018*). Hence Mcm2–Ctf4–Pol α forms an 'axis' for nucleosome transfer from parental DNA to the lagging daughter DNA (*Gan et al., 2018*). If only one Pol α-primase exists for two replication forks as in our 2CMG–Ctf4 factory model, during the time for transfer of a parental H3H4 to the lagging strand of one fork, the other fork in the opposite CMG would have progressed the same distance for transfer of another parental H3H4 to the leading strand. In this view, the architecture of our factory model facilitates the observed equal transfer of histones to both leading and lagging strand products (*Gan et al., 2018*; *Petryk et al., 2018*; *Yu et al., 2018*).

There is another aspect of the replisome factory architecture that may play a role in the proposed Mcm2-Ctf4-Pol α axis of nucleosome distribution. Two Mcm2's are needed to bind one H3H4 tetramer (*Huang et al., 2015*; *Richet et al., 2015*). In our structure, Mcm2 is ordered from Pro-201, and there is a 200 residue long peptide that is intrinsically disordered. So the H3H4 binding site (residues 61–130) is within this unstructured peptide. In our factory model, the two Mcm2's, one from each CMG helicase, may project out a pair of long tentacles (Mcm2 residues 1–200) between the two CMGs, and the two tentacles would conspire to capture one H3H4 to allow the 'Mcm–Ctf4–Pol α 'axis' to participate in equal deposition of parental histone marks on both strands of wt cells, much like the single Pol α-primase in the factory. An alternative to the use of two Mcm2's is that the H3H4 tetramer could be split so that one H3H4 dimer is bound by one Mcm2 and one Asf1 (*Huang et al., 2015*; *Richet et al., 2015*; *Wang et al., 2015*). However, mass spectrometry studies indicate that the H3H4 tetramer remains intact during replication (*Xu et al., 2010*).

Interestingly, Okazaki fragments are sized relative to the nucleosome repeat (*Smith and Whitehouse, 2012*). If nucleosome transfer is coordinated with Okazaki fragment synthesis, the parental nucleosome will occupy approximately the same sequence on the nascent DNA as it did on the parental DNA, as observed experimentally (*Madamba et al., 2017*). To conclude, the current study is only the beginning of a comprehensive understanding of how replication is organized in the cell. We expect that additional proteins, further layers of organization, and yet to be determined dynamics exist in nuclear replication factories. The replisome architecture and actions in the context of chromatin and epigenetic inheritance are important areas for future research.

## Materials and methods

### Reagents

Radioactive nucleotides were from Perkin Elmer and unlabeled nucleotides were from GE Healthcare. Protein concentrations were determined using the Bio-Rad Bradford Protein stain using BSA as a standard. Purification of *S. cerevisiae* Pol α-primase, Pol ε, CMG, the C-terminal half of Ctf4 (residues 471–927), and full length Ctf4 were purified according to previously published procedures (*Georgescu et al., 2014*; *Langston et al., 2014*). The C-terminal half of Ctf4 is necessary and sufficient for Ctf4 binding to CMG and Pol α-primase, as shown previously (*Simon et al., 2014*). Oligonucleotides were from IDT (Integrated DNA Technologies). ATPγS used for experiments in this report were purchased from Roche (catalog 11162306001). In experiments testing ATPγS from other companies, ATPγS was purchased from Sigma-Aldrich (catalog A1388) and Tocris Bioscience (catalog 4080).

### EM sample preparation of CMG–Ctf4$_3$ complexes

CMG (3.36 nmol) was mixed with Ctf4$_3$ (1.7 nmol) in a final volume of 1.37 mL of buffer A (20 mM Tris-acetate (pH 7.6), 1 mM DTT, 2 mM magnesium acetate) plus 200 mM KCl. The mixture was incubated on ice for 30 min, then injected onto a 0.25 mL MonoQ column equilibrated in buffer A + 200 mM KCl. The column was eluted with a 20-column linear gradient of buffer A + 200 mM KCl to buffer A + 600 mM KCl. Fractions of 0.25 mL were collected and protein concentration was determined using the Bio-Rad Bradford Protein stain and BSA as a standard. Fractions were analyzed in a Commasie stained 8% SDS-PAGE gel and gel lanes were scanned using a Typhoon 9400 laser imager (GE Healthcare). Scanned gels were analyzed using ImageQuant TL v2005 software and the two peak fractions were pooled and concentrated to 8.5 mg/ml in 79 μL. The sample was mixed with a 4-fold excess of a DNA 20-mer (5'-Cy3-dTdT$_{biotin}$dT$_{18}$) oligonucleotide, 0.2 mM AMPPNP

(final) which binds CMG. The sample was incubated 2 hr on ice then applied onto a Superose 6 Increase 3.2/300 gel filtration column (GE Healthcare) equilibrated in 20 mM Tris-acetate (pH 7.5), 1 mM DTT, 2 mM magnesium acetate, 60 mM potassium glutamate, 0.1 mM AMPPNP. The Cy3 DNA was added to visualize elution of CMG–Ctf4 at 565 nm, along with monitoring elution at 280 nM. Fractions were analyzed by SDS-PAGE and scanned on a Typhoon 9400 laser imager (GE Healthcare) to estimate the stoichiometry of CMG–to-Ctf4 in each fraction. The samples used for analysis were fraction 37 (0.35 mg/mL), fraction 35 (0.48 mg/mL), fraction 33 (0.42 mg/mL) and fraction 31 (0.35 mg/mL).

## Cryo-EM of Ctf4$_3$, Ctf4$_3$–Pol α-primase and Ctf4$_3$–CMG complexes

To prepare cryo-EM grids, individual fractions of CMG–Ctf4$_3$ from the gel filtration column were dialyzed against buffer A and concentrated to 0.35–0.48 mg/mL. Then 3 μL aliquots were applied to glow-discharged C-flat 1.2/1.3 holey carbon grids, incubated for 10 s at 6°C and 95% humidity, blotted for 3 s then plunged into liquid ethane using an FEI Vitrobot IV. In C-flat R1.2/1.3 holey carbon film grids, the CMG–Ctf4$_3$ particles distributed well for each of the samples. The Ctf4$_3$–CMG grids were loaded into an FEI Titan Krios electron microscope operated at a high tension of 300 kV and images were collected semi-automatically with EPU under low-dose mode at a nominal magnification of × 130,000 and a pixel size of 1.074 Å per pixel. A Gatan K2 summit direct electron detector was used under super-resolution mode for image recording with an under-focus ranging from 1.5 to 2.5 μm. A Bioquantum energy filter installed in front of the K2 detector was operated in the zero-energy-loss mode with an energy slit width of 20 eV. The dose rate was 10 electrons per Å$^2$ per second and total exposure time was 6 s. The total dose was divided into a 30-frame movie so each frame was exposed for 0.2 s. Approximately 5900 raw movie micrographs were collected.

Analysis of Ctf4$_3$ used a sample of Ctf4$_3$ at 1.8 mg/ml in 20 mM Tris-acetate (pH 7.5), 1 mM DTT, 2 mM magnesium acetate, 60 mM potassium glutamate. For the Pol α-primase–Ctf4 EM analysis we were unable to isolate a complex between these weak interacting components. Hence, we directly mixed the proteins for EM analysis. The samples contained Ctf4$_3$ and Polα-primase complex either at 1:1 molar ratio at a final concentration of 2.25 mg/ml, or at 1:3 molar ratio at a final concentration of 1.75 mg/ml, in 20 mM Tris-Acetate pH 7.5, 0.5 mM EDTA, 100 mM KGlutamate and incubated 20 min on ice. The samples were then applied to grids and plunge fronzen as described above. We collected ~ 1000 raw movie micrographs for each sample on an FEI Talos Arctica operated at 200 kV with a Falcon III direct electron detector. Data was collected semi-automatically with EPU at a nominal magnification of × 120,000 and pixel size of 1.21 Å per pixel. The under-focus value ranged from 1.5 to 2.5 μm. The dose rate was 20 e per Å$^2$ per second and total exposure time was 3 s. The total dose was divided into a 39-frame movie so each frame was exposed for 0.07 s.

## Negative-staining EM of Pol α-primase

A sample of 2 μL of yeast Pol α-primase at 0.1 mg/mL in 20 mM Tris-Acetate pH 7.5, 0.5 mM EDTA, 100 mM KGlutamate was applied to a glow-discharged carbon-coated copper grid for 30 s. Excess sample was blotted away using Whatman filter paper (Grade 1) with subsequent application of 2 μL of a 2% (w/v) aqueous solution of uranyl acetate. The staining solution was left on the grid for 30 s before the excess was blotted away. The staining solution was applied for a second time, blotted away and left to dry for 15 min before EM. A total of 50 micrographs were collected on a 2 k by 2 k CCD camera in FEI Tecnai G2 Spirit BioTWIN TEM operated at 120 kV at a magnification of × 30,000, corresponding to 2.14 Å per pixel at the sample level. Electron micrographs were processed using Relion 2 (*Scheres, 2012*). After CTF estimation and correction using Gctf (*Zhang, 2016*), particles in each micrograph were automatically picked using a Gaussian blob as the template at a threshold of 0.1, leading to a dataset of 24,712 raw particles. 2D classification was run for 25 iterations with the class number assigned to 100, the particle mask diameter set to 256 Å, the regularization parameter T left at the default value of 2, and the maximum number of significant coarse weights restricted to 500.

For observing the CMG–Ctf4$_3$–Pol α-primase complex, a sample of 2 μL of the ternary mixture at 0.12 mg/mL was applied to a glow-discharged carbon-coated grid for 1 min, then a 2 μL drop of 2% (w/v) aqueous solution of uranyl acetate was applied to the grid for and addtional 1 min, then blotted away with a piece of filter paper, and the staining step was repeated one more time. The dried

EM grid was loaded onto an 120 kV FEI Tecnai G2 Spirit EM with a 2K × 2K CCD camera. A total of 200 micrographs were collected at a magnification of × 30,000, corresponding to 2.14 Å per pixel. After CTF estimation and correction using CTFFIND4, particles were manually picked in Relion 2.1. About 9000 particles were picked for 2D classification. Several 2D averages showed a binary complex of CMG–Ctf4$_3$, and the ternary complex of CMG–Ctf4$_3$–Pol α-primase complex.

## Image processing and 3D reconstruction

The movie frames were first aligned and superimposed by the program Motioncorr 2.0 (*Zheng et al., 2017*). Contrast transfer function parameters of each aligned micrograph were calculated using the program CTFFIND4 (*Rohou and Grigorieff, 2015*). All the remaining steps, including particle auto selection, 2D classification, 3D classification, 3D refinement, and density map postprocessing were performed using Relion-2.1 (*Scheres, 2012*). For the 1CMG–Ctf4$_3$ sample, templates for automatic picking were generated from 2D averages calculated from about ~ 10,000 manually picked particles in different views. Automatic particle selection was then performed for the entire data set, and 759,267 particles were picked. Selected particles were carefully inspected; 'bad' particles were removed, some initially missed 'good' particles were re-picked, and the remaining good particles were sorted by similarity to the 2D references, in which the bottom 10% of particles with the lowest z-scores were removed from the particle pool. 2D classification of all good particles was performed and particles in the classes with unrecognizable features by visual inspection were removed. A total of 564,011 particles were then divided into two groups containing either a single CMG or two or three copies of CMG for further 3D classification. For 3D reconstruction of the various Ctf4$_3$–CMG complexes, the low pass-filtered CMG-apo structure was used as the starting model, leading to the first 3D map of the Ctf4$_3$–(CMG)$_3$ complex. We then used Chimera to mask out either one or two CMG densities from the 3D map of Ctf4$_3$–(CMG)$_3$ to generate a starting model for 1CMG–Ctf4$_3$ and 2CMG–Ctf4$_3$ respectively. Four 3D models were derived from each group, and models that appeared similar were combined for final refinement. The models not chosen were distorted and those particles were discarded. The final three datasets that contain single CMG (Ctf4$_3$–CMG$_1$), two CMGs (Ctf4$_3$–CMG$_2$), and three CMGs (Ctf4$_3$–CMG$_3$) were used for final 3D refinement, resulting in three 3D density maps at 3.9 Å, 5.8 Å and 7.0 Å resolution, respectively. The resolution was estimated by the gold-standard Fourier shell correlation, at the correlation cutoff value of 0.143. The 3D density map was sharpened by applying a negative B-factor of −146, −135 and −143 Å$^2$, respectively. Local resolution was estimated using ResMap. In the 1CMG–Ctf4$_3$ analysis, the density of the Mcm2-7 CTD motor region was weak and noisy. A mask was used to exclude the CTD ring, and the remaining region composed of Ctf4$_3$–Cdc45−GINS−Mcm2-7 N-tier ring had an estimated resolution of 3.8 Å, based on the gold standard Fourier shell correlation curve.

For the Ctf4$_3$–Polα-primase dataset, ~5000 particles in different views were manually picked to generate the templates. Automatic particle selection was then performed for the entire dataset, and 237,688 particles were initially picked in Relion-2.1. Similar to the image process in Ctf4$_3$–CMG, 'bad' and structurally heterogeneous particles were removed by visual inspection. After 2D classification, 142,806 particles in many different views were selected for further 3D classification. The selected particles were separated into five classes by the 3D classification procedure, using a low-pass filtered Ctf4 trimer structure as the starting model. Two good classes were selected and combined for final 3D refinement. The final 3D refinement produced a 12 Å 3D density map. The resolution of the map was estimated by the gold-standard Fourier shell correlation, at the correlation cutoff value of 0.143. The 3D density map was sharpened by applying a negative B-factor of −162 Å$^2$.

For the Ctf4$_3$ dataset, we picked about 3000 particles in different views to generate the templates. Automatic particle picking was performed for the entire dataset containing about 500 micrographs, and 113,936 particles were picked in Relion-2.1; then 2D classification was performed to produce a set of well-defined 2D averages. 3D classification and 3D reconstruction were not performed because the crystal structure of Ctf4$_3$ is available and our purpose was only to obtain the cryo-EM 2D averages.

## Atomic modeling, refinement, and validation

The modeling of Ctf4$_3$–CMG$_1$, Ctf4$_3$–CMG$_2$ and Ctf4$_3$–CMG$_3$ was based on the structure of CMG (PDB ID 3JC5) and Ctf4$_3$ (PDB 4C8H). For Ctf4$_3$–CMG$_1$, one CMG and one Ctf4 trimer were directly docked as rigid bodies into the EM map using Chimera (*Pettersen et al., 2004*). The initial modeling was followed by further manual adjustments using COOT (*Emsley et al., 2010*), guided by residues with bulky side chains like Arg, Phe, Tyr and Trp. The improved model was then refined in real space against the EM densities using the phenix.real_space_refine module in PHENIX (*Adams et al., 2010*). For Ctf4$_3$–CMG$_2$ and Ctf4$_3$–CMG$_3$, Chimera was used to rigid-body dock two or three copies of CMG and one Ctf4$_3$ into the corresponding EM map. Due to the low resolution of the latter two maps, the atomic models were not subject to further refinement. Finally, the quality of the refined atomic model of Ctf4$_3$–CMG was examined using MolProbity (*Chen et al., 2010*). For modeling of the Ctf4$_3$–Polα-primase 3D map, one Ctf4 trimer (PDB 4C8H) and one Polα-primase catalytic NTD (PDB ID 4FYD) were docked as two separate rigid bodies into the 3D map in Chimera. Due to the low resolution of 3D map, the model was neither manually adjusted nor subjected to refinement. Structural figures were prepared in Chimera and Pymol (https://www.pymol.org).

## Glycerol gradient sedimentation

To examine complex formation, 0.12 nmol of CMG was mixed with 0.12 nmol Ctf4$_3$, 0.12 nmol of Pol ε and 0.12 nmol Pol α-primase in 150 µl of 20 mM Tris-acetate (pH 7.6), 1 mM DTT, 2 mM Mg-OAc, 100 mM NaCl (final) for 30 min at 16°C, and the mixture was layered on top of an 11.2 mL 15–35% (vol/vol) glycerol gradient in 20 mM Tris-acetate (pH 7.6), 1 mM DTT, 2 mM magnesium acetate, 50 mM NaCl and spun at 4°C for 16 hr in a Sorvall 90SE ultracentrifuge using a T-865 rotor. Five drop fractions were collected from the bottom of the centrifuge tubes, and 20 µL samples were analyzed by SDS-PAGE stained with Coomassie Blue. Similar gradient analyses were performed for sub-mixtures of the proteins. A parallel gradient was also performed using protein standards (BioRad 151–1901) in the same buffer. Gels were scanned and quantitated using Image J software and relative moles of protein in each band were calculated, taking into account their native mw. Only subunits the size of Cdc45 or larger were analyzed, as smaller subunit bands were too light for analysis. The full-length Ctf4$_3$ was used to enhance its staining capacity, as the small C-half of Ctf4$_3$ was not well distinguished above background. The full-length Ctf4$_3$ overlapped with Mcm4 and required taking the overlap into account (see legend to *Figure 7—figure supplement 1*).

## Pull-down assays

Pull-down assays were performed by mixing 20 pmol of N-terminal labeled streptag-Ctf4 trimer with 60 pmol Pol α-primase (when present) and 60 pmol CMG (when present) at 25°C for 10 min. Then the volume of the protein solution was adjusted to 50 µl with binding buffer (20 mM Tris-Acetate, pH 7.5, 1 mM DTT, 5% glycerol, 2 mM magnesium acetate, 50 mM KGlutamate and 50 mM KCl) before being mixed with 50 µL of a 50% suspension of StrepTactin magnetic beads (Qiagen). The protein-bead mixture was incubated in a Thermomixer at 1250 rpm, 4°C for 1 hr. The beads were then collected with a magnetic separator and the supernatant containing unbound proteins was removed. The beads were washed twice with 100 µL binding buffer and bound proteins were eluted by incubating the beads in 50 µL of the same buffer supplemented with 5 mM biotin at 25°C for 15 min. The eluted proteins were analyzed in an 8% SDS-PAGE gel and scanned and quantitated using a Typhoon 9400 laser imager (GE Healthcare).

## Helicase assays

DNA oligos used to form the forked substrate having a 60 bp duplex stem were 'leading helicase oligo': 5'-GGCTCGTTTTACAACGTCGTGCTGAGGTGATATCTGCTGAGGCAATGGGAA TTCGCCAACCTTTTTTTTTTTTTTTTTTTTTTTTTTTTTTTTTTTTTTTT*T*T*T 3', and lagging helicase oligo: 5'-GGCAGGCAGGCAGGCAGGCAGGCAGGCAGGCAGGCAGGCAGGTTGGCGAATTCCCA TTGCCTCAGCAGATATCACCTCAGCACGACGTTGTAAAAC*G*A*G-3'. These oligos were identical (Oligo-2) or slightly modified (Oligo-1) from those used in *Kose et al. (2019)*. Asterisks denote thiodiester linkages. The leading helicase oligo was $^{32}$P-5' end-labeled and annealed to its respective lagging helicase oligo to form the forked DNA substrate. The forked DNA was isolated from a native PAGE. In the experiments of *Figure 7b*, 2CMG–Ctf4$_3$ complexes were formed by preincubating 276

nM CMG with 138 nM full length Ctf4$_3$ on ice for 15 min. and then 25°C for 10 min. CMG or CMG–Ctf4 complexes were added to reactions to give a final concentration of 24 nM CMG, 0.5 nM forked DNA substrate, 5 mM ATP and, when present, 12 nM Ctf4 (as trimer) in a final buffer of 20 mM Tris-Acetate pH 7.6, 5 mM DTT, 0.1 mM EDTA, 10 mM MgSO$_4$, 30 mM KCl. Reactions were preincubated with 0.1 mM ATPγS (Roche) for 120 min at 30°C with DNA and either CMG or CMG–Ctf4 before initiating unwinding upon adding 5 mM ATP. Reactions were quenched at the times indicated with 20 mM EDTA and 0.1% SDS (final concentrations), and analyzed on a 10% native PAGE in TBE buffer. The assays of *Figure 7—figure supplement 3* (i.e. that tested preincubation time with 0.1 mM ATPγS) were performed as in *Figure 7b*, except reactions were preincubated for different times with ATPγS (Roche) at 30°C before initiating unwinding with 5 mM ATP. Gels were exposed to a phosphor screen then scanned and quantitated on a Typhoon 9400 laser imager (GE Healthcare).

## Replication assays

Leading strand replication experiments used a forked DNA substrate having a 60 bp duplex stem formed from the following oligonucleotides: Leading strand template 180mer: 5'-AGGTGTAGATTAATGTGGTTAGAATAGGGATGTGGTAGGAAGTGAGAATTGGAGAGTGTG TTTTTTTTTTTTTTTTTTTTTTTTTTTTTTTTTTTTTTTAAAGGTGAGGGTTGGGAAGTGGAAGGA TGGGCTCGAGAGGTTTTTTTTTTTTTTTTTTTTTTTTTTTTTTTTTTTTTT*T*T*T-3', and lagging strand template 100mer: 5'-GGCAGGCAGGCAGGCAGGCAGGCAGGCAGGCAGGCAGGCACACACTC TCCAATTCTCACTTCCTACCACATCCCTATTCTAACCACATTAATCTACA*C*C*T-3'. Asterisks are thiodiester linkages. The fork was primed for leading strand DNA replication with a DNA 37mer (5'-32P-CCTCTCGAGCCCATCCTTCCACTTCCCAACCCTCACC-3'). The CMG– Ctf4$_3$ complexes were reconstituted by incubating 276 nM CMG with 138 nM Ctf4$_3$ on ice for 15 min. and then 25°C for 10 min. For the reactions of *Figure 7c*, reactions contained 20 nM of CMG, 10 nM Ctf4$_3$ (where indicated) 0.5 nM $^{32}$P-primed forked substrate (final concentrations) in a buffer consisting of 25 mM Tris-Acetate pH 7.5, 5% glycerol, 2 mM DTT, 10 mM magnesium sulfate, 1 μM dTTP and preincubated with 100 μM ATPγS for 115 min, then 20 nM Pol α-primase was added and 5 min later DNA synthesis was initiated upon adding 5 mM ATP and 100 μM each dNTP. Reactions were stopped at the indicated times by removing aliquots and adding an equal volume of 1% SDS, 40 mM EDTA, 90% formamide. For the preincubation analysis of *Figure 7—figure supplement 4*, the same procedure was followed as *Figure 7c*, except the preinubation time with Pol α-primase varied as indicated in the figure. Reaction products were analyzed on 10% PAGE gels containing 6M urea in TBE buffer, then exposed to a phosphorimager screen and imaged with a Typhoon FLA 9500 (GE Healthcare). Gel bands were quantitated with ImageQuant software.

## Accession codes

The 3D cryo-EM maps of Ctf4$_3$–CMG$_1$, Ctf4$_3$–CMG$_2$, and Ctf4$_3$–CMG$_3$ at 3.8 Å, 5.8 Å and 7.0 Å resolution have been deposited in the Electron Microscopy Data Bank under accession codes EMD-20471, EMD-20472 and EMD-20473, respectively. The corresponding atomic models have been deposited in the Protein Data Bank under accession codes 6PTJ, 6PTN, and 6PTO, respectively.

## Acknowledgements

Cryo-EM data were collected at the David Van Andel Advanced Cryo-Electron Microscopy Suite at the Van Andel Institute. We thank Xing Meng for help with data collection. This study was supported by the US National Institutes of Health grants GM131754 (to HL) and GM115809 (to MEO), and the Howard Hughes Medical Institute (MEO).

## Additional information

### Funding

| Funder | Grant reference number | Author |
| --- | --- | --- |
| National Institutes of Health | GM115809 | Michael E O'Donnell |
| National Institutes of Health | GM131754 | Huilin Li |

| Howard Hughes Medical Institute | Michael E O'Donnell |

The funders had no role in study design, data collection and interpretation, or the decision to submit the work for publication.

## Author contributions
Zuanning Yuan, Data curation, Formal analysis, Validation, Investigation, Visualization, Methodology, Writing—original draft, Writing—review and editing; Roxana Georgescu, Nina Y Yao, Resources, Formal analysis, Investigation, Methodology; Ruda de Luna Almeida Santos, Investigation, Visualization; Daniel Zhang, Resources, Data curation, Investigation, Methodology; Lin Bai, Data curation, Formal analysis, Validation, Investigation, Visualization, Methodology; Gongpu Zhao, Data curation, Formal analysis, Visualization, Methodology; Michael E O'Donnell, Huilin Li, Conceptualization, Resources, Data curation, Formal analysis, Supervision, Funding acquisition, Validation, Investigation, Visualization, Methodology, Writing—original draft, Project administration, Writing—review and editing

## Author ORCIDs
Roxana Georgescu (iD) http://orcid.org/0000-0002-1882-2358
Michael E O'Donnell (iD) https://orcid.org/0000-0001-9002-4214
Huilin Li (iD) https://orcid.org/0000-0001-8085-8928

## Decision letter and Author response
Decision letter https://doi.org/10.7554/eLife.47405.045
Author response https://doi.org/10.7554/eLife.47405.046

# Additional files

## Supplementary files
• Transparent reporting form DOI: https://doi.org/10.7554/eLife.47405.031

## Data availability
The 3D cryo-EM maps of $Ctf4_3$-$CMG_1$, $Ctf4_3$-$CMG_2$, and $Ctf4_3$-$CMG_3$ at 3.8-Å, 5.8-Å and 7.0-Å resolution have been deposited in the Electron Microscopy Data Bank under accession codes EMD-20471, EMD-20472 and EMD-20473, respectively. The corresponding atomic models have been deposited in the Protein Data Bank under accession codes PDB 6PTJ, PDB 6PTN, PDB 6PTO, respectively.

The following datasets were generated:

| Author(s) | Year | Dataset title | Dataset URL | Database and Identifier |
|---|---|---|---|---|
| Yuan Z, Georgescu R, de Luna Almeida Santos R, Zhang D, Bai L, Yao NY, Zhao G, O'Donnell ME, Li H | 2019 | 3D cryo-EM map of $Ctf4_3$-$CMG_1$ | https://www.ebi.ac.uk/pdbe/emdb/EMD-20471 | Electron Microscopy Data Bank, EMD-20471 |
| Yuan Z, Georgescu R, de Luna Almeida Santos R, Zhang D, Bai L, Yao NY, Zhao G, O'Donnell ME, Li H | 2019 | 3D cryo-EM map of $Ctf4_3$-$CMG_2$ | https://www.ebi.ac.uk/pdbe/emdb/EMD-20472 | Electron Microscopy Data Bank, EMD-20472 |
| Yuan Z, Georgescu R, de Luna Almeida Santos R, Zhang D, Bai L, Yao NY, Zhao G, O'Donnell ME, Li H | 2019 | 3D cryo-EM map of $Ctf4_3$-$CMG_3$ | https://www.ebi.ac.uk/pdbe/emdb/EMD-20473 | Electron Microscopy Data Bank, EMD-20473 |

| Yuan Z, Georgescu R, de Luna Almeida Santos R, Zhang D, Bai L, Yao NY, Zhao G, O'Donnell ME, Li H | 2019 | Atomic model of Ctf4$_3$-CMG$_1$ | https://www.rcsb.org/structure/6PTJ | Protein Data Bank, 6PTJ |
|---|---|---|---|---|
| Yuan Z, Georgescu R, de Luna Almeida Santos R, Zhang D, Bai L, Yao NY, Zhao G, O'Donnell ME, Li H | 2019 | Atomic model of Ctf4$_3$-CMG$_2$ | https://www.rcsb.org/structure/6PTN | Protein Data Bank, 6PTN |
| Yuan Z, Georgescu R, de Luna Almeida Santos R, Zhang D, Bai L, Yao NY, Zhao G, O'Donnell ME, Li H | 2019 | Atomic model of Ctf4$_3$-CMG$_3$ | https://www.rcsb.org/structure/6PTO | Protein Data Bank, 6PTO |

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
