## [Decision Letter]

Thank you for submitting your article "Ctf4 organizes sister replisomes and Pol α into a replication factory" for consideration by *eLife*. Your article has been reviewed by two peer reviewers, and the evaluation has been overseen by a Reviewing Editor and Cynthia Wolberger as the Senior Editor. The reviewers have opted to remain anonymous.

The reviewers have discussed the reviews with one another and the Reviewing Editor has drafted this decision to help you prepare a revised submission.

Summary:

The current work presents the 3.8 Å resolution structure of a CMG complex bound to Ctf4. The structures for Ctf4 occupied by 2 or 3 CMG complexes (at 6Å and 7 Å resolution) are also shown, along with a model Ctf4-Pol-α/primase (obtained at 12 Å resolution) that identifies one lone tethering point between Pol-α and Ctf4. Based on the ability of Ctf4 to bind multiple CMGs and a single Pol-α, a model for how Ctf4 coordinates the action of two helicases and the lagging strand polymerase in a prospective replication factory is proposed. This framework is then used to speculate as to how factories might promote sister chromosome cohesion.

The strength of this manuscript lies in the structural studies of various Ctf4 complexes. This information is important as it identifies two previously unknown Ctf4 interactions with CMG subunits Cdc45 and Psf2 and substantiates the prediction that multiple CMG assemblies can bind to Ctf4 without clashing. In addition, Ctf4 is found to preferentially bind one DNA Pol-α/primase. Overall, these insights contribute to our general understanding of the architecture of the eukaryotic replisome. This said, there are several extant issues that need to be addressed.

Essential revisions:

1) A substantial portion of the paper is dedicated to several elaborate models that are consistent with available data but never explicitly tested, such as the idea that replication factories might have a role in facilitating sister chromatid cohesion. The work would be substantially improved if mutations in the newly discovered Ctf4-CMG interface (or even a Ctf4 deletion, which is viable) could be examined for their ability to disrupt the observation made by Tanaka and colleagues showing that DNA at equal distances from an origin come together at the time of replication. Otherwise discussion topics such as the role of Pol-α-Ctf4-CMG in parental histone re-deposition appear more solidly grounded. Barring the addition of genetic or cell biological studies aimed at testing aspects of the cohesion model, it would be appropriate to focus the discussion on the Ctf4-CMG interaction (i.e., eliminating the speculation about cohesion and Figure 8—figure supplement 2B) and to limit the Discussion to the two-CMG-one Pol-α/primase model.

2) The manuscript does not take into account that amino acid changes in the CIP motif in Chl1 (a cohesin interactor) give rise to a sister chromatid cohesion phenotype, unlike a mutation in the CIP motif of Sld5. Taken at face value, the evidence indicates that Chl1 interaction with Ctf4, and not two CMGs bridged by Ctf4, is important for sister chromatid cohesion. If genetic or cell biological data are presented to test the proposed cohesion model, these findings will need to be noted.

3) The Pol-α modeling is based on less solid data than the higher-resolution CMG–Ctf4 interactions. There is some concern as to veracity of the placement the different Pol-α/primase elements with respect to Ctf4 and/or the CMG. Since the location of these regions is an important part of the replication factory model, it is important that more data be shown to boost confidence about their placement. Details of the cryo-EM volume with different views/different thresholds should be shown, as well as the model-to-map Fourier Shell Correlation plot comparing best and second-best docking solution. Was RELION multi-body refinement attempted for this complex? The resolution of the Pol-α lobe might improve using this strategy. Also, panels A and B/C of Figure 6 should be scaled so that the Pol-α NTD has the same size.

4) In the Sun, 2015, replisome paper (Figure 5E), Pol-α/primase maps to the opposite Ctf4 side compared to that described in the current study. This discrepancy should be addressed more clearly.

5) The biochemical data in Figures S13, S14, and S15 are somewhat buried, which detracts from the impact of the work. It would be preferable for these panels to be included in the main results in place of the extensive discussion of the model that is there now. Along these lines, is this the first time that the binding of the CMG to Ctf4 is seen to stimulate Pol-α binding (i.e., that their association is cooperative) and vice versa? If so, please note as much, or else cite the appropriate references that the findings confirm. Similarly, is this first time that stoichiometric studies have been performed to show that helicase activity is supported by a dual-CMG/Ctf4 complex? Do the authors think that the difference between 1.7 CMG–Ctf4-3 versus 1 CMG–Ctf4-3 in Figure S15 is a result of cooperative action of the two helicases? If so, how might this cooperativity emerge?

6) It would seem that only the double CMG–Ctf4-3 works well in the elongation assay (in S15B), which contradicts the idea that one or two CMGs bound to Ctf4 bind and unwind as well as CMG alone (S14). Please describe the tests that were performed to determine whether the time of preincubation was long enough to ensure that only unwinding is being measured in these assays rather than fork binding and unwinding. Based on recent work from the Yardimci lab, it would seem that many previous studies are likely to be measuring binding as well as unwinding (with binding being rate limiting).

[Editors' note: further revisions were requested prior to acceptance, as described below.]

Thank you for submitting your article "Ctf4 organizes sister replisomes and Pol α into a replication factory" for consideration by *eLife*. Your article has been reviewed by two peer reviewers, and the evaluation has been overseen by a Reviewing Editor and Cynthia Wolberger as the Senior Editor. The reviewers have opted to remain anonymous.

The reviewers have discussed the reviews with one another and the Reviewing Editor has drafted this decision to help you prepare a revised submission.

Primary issues:

1) In the prior review, it was requested that "Details of the cryo-EM volume with different views/different thresholds should be shown, as well as the model-to-map Fourier Shell Correlation plot comparing best and second-best docking solution." The reply states that "We did not provide a map-to-coordinated correction curve, because we simply used rigid body docking – the original crystal structures were unaltered." This reply is unclear. Do the authors mean "map-to-coordinate correlation curve"?

It was requested that a map-to-model correlation curve be shown for the best rigid body docking solution and the second-best rigid body docking solution. If the cryo-EM map is sufficiently featured to model one Pol α orientation with confidence, then the FSC for the best solution should show significantly higher correlation than the second-best solution. The authors could also simply quote the average correlation score for the best vs second best docking solution. Please address.

2) The evidence that the CMG helicase can use ATPγS to unwind DNA is not compelling. The data in Figure 7B appear to use extremely high concentrations of nucleotide (100 and 500 mM) concentrations – are the units actually μM? The ATPγS may be contaminated with ATP (which is commonly the case), and it may be this nucleotide rather than ATPγS that is used by the CMG.

The more relevant data to this point may be in Figure 7—figure supplement 3, where the experiment is performed at a more realistic concentration of 0.1 mM ATPγS. Here it is suggested that the same amount of unwinding is seen with ATPγS as when ATP is added. There are two issues with these data. First, in the ATPγS data, the overall signal is weak and the detected released DNA is very weak. In contrast, the experiment with ATP shows a much more robust signal. Second, there is no direct comparison with ATP versus ATPγS (as in 7B). Experiments using equivalent concentrations of ATPγS and ATP at similar (low) concentrations need to be shown to provide a direct comparison.

3) Related to (2), an important issue that is not addressed is whether there is a very small amount of a contaminating helicase in the preps that is responsible for the unwinding. Given the very slow different kinetics seen with ATPγS (it takes around 30 minutes and then makes up all of the slower rates observed earlier) and previous data from others (where the hydrolysis rate found for Mcm2-7 with ATPγS was 2% of that seen with ATP), it seems possible that the observed activity derives from a contaminant. A mutant form of the CMG carrying a critical ATPase motif substitution that is necessary for unwinding needs to be tested to demonstrate that unwinding is indeed lost.

Alternatively, the ATPγS data and claims could just be removed, as they are not central to the primary thrust of the paper.

---

## [Author Response]

Essential revisions:1) A substantial portion of the paper is dedicated to several elaborate models that are consistent with available data but never explicitly tested, such as the idea that replication factories might have a role in facilitating sister chromatid cohesion. The work would be substantially improved if mutations in the newly discovered Ctf4-CMG interface (or even a Ctf4 deletion, which is viable) could be examined for their ability to disrupt the observation made by Tanaka and colleagues showing that DNA at equal distances from an origin come together at the time of replication. Otherwise discussion topics such as the role of Pol-α-Ctf4-CMG in parental histone re-deposition appear more solidly grounded. Barring the addition of genetic or cell biological studies aimed at testing aspects of the cohesion model, it would be appropriate to focus the discussion on the Ctf4-CMG interaction (i.e., eliminating the speculation about cohesion and Figure 8—figure supplement 2B) and to limit the Discussion to the two-CMG-one Pol-α/primase model.

We agree with the reviewers and thank them for this comment. The suggested “Tanaka” microscope experiments were, in fact, already underway +/- a Ctf4 deletion, but it will take time and we plan this to be a follow-up study along with some new EM studies that we plan to perform for this follow-up as well. Hence, we have taken the reviewer’s second suggestion and removed proposals on cohesion, and limited the discussion to the CMG–Ctf4 interaction, the proposed functioning of the 2CMG-1-Pol α/primase model, and speculation about a possible role in parental histone re-deposition on nascent DNA products as the reviewers suggest.

2) The manuscript does not take into account that amino acid changes in the CIP motif in Chl1 (a cohesin interactor) give rise to a sister chromatid cohesion phenotype, unlike a mutation in the CIP motif of Sld5. Taken at face value, the evidence indicates that Chl1 interaction with Ctf4, and not two CMGs bridged by Ctf4, is important for sister chromatid cohesion. If genetic or cell biological data are presented to test the proposed cohesion model, these findings will need to be noted.

As with comment 1, we no longer propose a role of the factory in cohesion, and instead take the reviewer’s suggestion of discussing how the structures observed here may relate to parental nucleosome transfer.

3) The Pol-α modeling is based on less solid data than the higher-resolution CMG–Ctf4 interactions. There is some concern as to veracity of the placement the different Pol-α/primase elements with respect to Ctf4 and/or the CMG. Since the location of these regions is an important part of the replication factory model, it is important that more data be shown to boost confidence about their placement. Details of the cryo-EM volume with different views/different thresholds should be shown, as well as the model-to-map Fourier Shell Correlation plot comparing best and second-best docking solution. Was RELION multi-body refinement attempted for this complex? The resolution of the Pol-α lobe might improve using this strategy. Also, panels A and B/C of Figure 6 should be scaled so that the Pol-α NTD has the same size.

For the Ctf4-Pol α 3D reconstruction: we have added a Euler angle distribution and the FSC resolution curve to the supplemental figure. We have also added in panel b the surface view in two thresholds – low and high levels. We did not provide a map-to-coordinated correction curve, because we simply used rigid body docking – the original crystal structures were unaltered.

For the Figure 4 CMG–Ctf4 interface – we have removed the square boxes as requested.

In Figure 6, we have re-scaled the panel a, such that panels A, B and C are in the same scale now.

4) In the Sun, 2015, replisome paper (Figure 5E), Pol-α/primase maps to the opposite Ctf4 side compared to that described in the current study. This discrepancy should be addressed more clearly.

We appreciate this comment and have clarified that the observation in the two reports are compatible, based on the Pol α-primase bilobed flexible structure, and presence of CMG/DNA in one study and lack of CMG/DNA in the other study. Specifically, Pol α-primase is an elongated particle with a Pol lobe on one end and primase lobe on the other, with up to 70o flexing between them (as cited). The structure of Pol α-primase-Ctf4 in this report does not include CMG or DNA, and thus only observes the direct Pol1-Ctf4 interaction, the primase lobe cannot be seen likely due to the inherent flexibility between Pol1 and primase lobes. In contrast, our previous study included CMG and a forked DNA, along with Ctf4 and Pol α-primase. Given the documented primase- Mcm interaction (as cited) and the presence of forked DNA in the earlier study, the observed density was likely the primase lobe of Pol α-primase. Thus the two studies likely observe different lobes of the flexible Pol α-primase. This is now described and clarified in the revised manuscript.

5) The biochemical data in Figures S13, S14, and S15 are somewhat buried, which detracts from the impact of the work. It would be preferable for these panels to be included in the main results in place of the extensive discussion of the model that is there now. Along these lines, is this the first time that the binding of the CMG to Ctf4 is seen to stimulate Pol-α binding (i.e., that their association is cooperative) and vice versa? If so, please note as much, or else cite the appropriate references that the findings confirm. Similarly, is this first time that stoichiometric studies have been performed to show that helicase activity is supported by a dual-CMG/Ctf4 complex? Do the authors think that the difference between 1.7 CMG–Ctf4-3 versus 1 CMG–Ctf4-3 in Figure S15 is a result of cooperative action of the two helicases? If so, how might this cooperativity emerge?

We appreciate this comment. Yes, this is the first demonstration of cooperativity in binding of Pol α-primase/CMG to Ctf43, and the first time that helicase and replication activity is conducted within the context of a 2 CMG–Ctf4 complex vs a single CMG. We now provide a main text figure (new Figure 7) of this biochemical data, and we performed new helicase and replication assays in triplicate +/- Ctf4 for this figure (panels C,D). The replication data using SEC fractions of pre-formed CMG–Ctf4 complexes used 20nM Ctf4 from each column fraction, and thus different amounts of CMG were present in each reaction resulting in the higher activity of fractions having ratios of 1.7 CMG:1 Ctf4 vs 1.0 CMG:1Ctf4. This figure is retained as a supplement figure and the use of constant 20 nM Ctf4 (and thus different amounts of CMG) is now clarified in text and legend of the revised manuscript.

We have added to this new main text figure our finding that ATPγS supports CMG unwinding (panel b). We tested ATPγS from 3 different manufactures, including Roche, and obtain the same result (in Figure 7, panel B). We discovered this while addressing main point #6, below.

6) It would seem that only the double CMG–Ctf4-3 works well in the elongation assay (in S15b), which contradicts the idea that one or two CMGs bound to Ctf4 bind and unwind as well as CMG alone (S14). Please describe the tests that were performed to determine whether the time of preincubation was long enough to ensure that only unwinding is being measured in these assays rather than fork binding and unwinding. Based on recent work from the Yardimci lab, it would seem that many previous studies are likely to be measuring binding as well as unwinding (with binding being rate limiting).

We appreciate this comment, and have addressed the first sentence in comment #5. Regarding whether preincubation time was long enough for CMG binding, we initially did not try to separate DNA binding and unwinding. We have changed that, and have performed the requested preincubation experiments using ATPγS, the typical nucleotide analogue the field (and Yardimci) uses for CMG/DNA preincubation for CMG binding DNA prior to adding ATP. We show a study of the preincubation times for helicase and replication assays in supplemental figures, as requested – the conclusion remains the same (i.e. Ctf4 does not increase CMG activity). The two recent Yardimchi reports that the reviewer’s mention use 0.1-0.5 mM ATPγS and preincubate 2h at 37°C, but neither report shows the result of this preincubation, nor do these reports document that this amount of time is required for CMG DNA binding. The one supplemental fluorescent figure suggests a lag of 10 min, and that this is the time of CMG binding. We do not know why 2 hours, 37°C was used as we have not been able to locate the rationale for this in Yardimchi’s reports. However, in doing the requested preincubation experiments, we surprisingly found that ATPγS supports CMG unwinding, and 2h and 30°C gives about 50% unwinding, the maximum observed in the Yardimchi work and our own. Therefore we now document here, for the first time we are aware of, that ATPγS in fact supports CMG unwinding of DNA (we used ATPγS from the same catalogue number and supplier as Yardimchi). We also tried ATPγS from two other companies, with the same result. We also tested 3 different CMG preps with the same result.

Given the request by reviewers to use similar conditions as Yardimchi, we found that use of 10 min at 0.1 mM ATPγS preincubation is sufficient for binding in helicase assays, and is below the time of observable unwinding, (see supplemental figures to Figure 7). For CMG to bind the long 3’ tail of the replication fork, which is also primed with an oligonucleotide, we needed a 30 min preincubation of CMG with ATPγS (Figure 7—figure supplement 5). The results, after the preincubation, continue to support that two CMGs on a Ctf4 trimer are active, and thus capable of function within a factory.

We have also retained the original supplemental helicase and replication assays as part b of their respective supplemental figures that document the preincubation time controls. Thus, even when both binding and unwinding/replication are measured, the CMG and CMG multimerized on Ctf4 have similar activity.

[Editors' note: further revisions were requested prior to acceptance, as described below.]

Primary issues:1) In the prior review, it was requested that "Details of the cryo-EM volume with different views/different thresholds should be shown, as well as the model-to-map Fourier Shell Correlation plot comparing best and second-best docking solution." The reply states that "We did not provide a map-to-coordinated correction curve, because we simply used rigid body docking – the original crystal structures were unaltered." This reply is unclear. Do the authors mean "map-to-coordinate correlation curve"?It was requested that a map-to-model correlation curve be shown for the best rigid body docking solution and the second-best rigid body docking solution. If the cryo-EM map is sufficiently featured to model one Pol α orientation with confidence, then the FSC for the best solution should show significantly higher correlation than the second-best solution. The authors could also simply quote the average correlation score for the best vs second best docking solution. Please address.

We have added the model-to-map correlation curves for Ctf4-CMG dimer, and Ctf4-CMG trimer in their appropriate supplemental figures (ie. Figure 5—figure, supplements 1 and 2). The revised manuscript also includes the map-to-model correlation curves for the best rigid body docking solution and the second-best rigid body docking solution for Pol α with Ctf4 (Figure 6—figure supplement 1).

2) The evidence that the CMG helicase can use ATPγS to unwind DNA is not compelling. The data in Figure 7B appear to use extremely high concentrations of nucleotide (100 and 500 mM) concentrations – are the units actually μM? The ATPγS may be contaminated with ATP (which is commonly the case), and it may be this nucleotide rather than ATPγS that is used by the CMG.The more relevant data to this point may be in Figure 7—figure supplement 3, where the experiment is performed at a more realistic concentration of 0.1 mM ATPγS. Here it is suggested that the same amount of unwinding is seen with ATPγS as when ATP is added. There are two issues with these data. First, in the ATPγS data, the overall signal is weak and the detected released DNA is very weak. In contrast, the experiment with ATP shows a much more robust signal. Second, there is no direct comparison with ATP versus ATPγS (as in 7b). Experiments using equivalent concentrations of ATPγS and ATP at similar (low) concentrations need to be shown to provide a direct comparison.

We appreciate this comment – and have dropped the ATPγS unwinding experiments from the revision, as requested by the reviewers. While we now know that CMG (but not Mcm2-7) can, in fact, hydrolyze ATPγS and couple it to unwind DNA (supported by the requested mutation analysis and by other means), we find it is prevented by certain sequences. Hence, we used sequences that are not unwound by CMG with ATPγS to repeat the preincubation, unwinding and replication experiments for this paper (Figure 7 and Figure 7—supplements 3, 4) and omit the earlier ATPγS work from the revised paper.

As the reviewers intuited, we had mislabeled Figure 7. The amount of ATPγS was μM not millimolar. ATPγS has no contaminating ATP due to the process in which it is synthesized.

3) Related to (2), an important issue that is not addressed is whether there is a very small amount of a contaminating helicase in the preps that is responsible for the unwinding. Given the very slow different kinetics seen with ATPγS (it takes around 30 minutes and then makes up all of the slower rates observed earlier) and previous data from others (where the hydrolysis rate found for Mcm2-7 with ATPγS was 2% of that seen with ATP), it seems possible that the observed activity derives from a contaminant. A mutant form of the CMG carrying a critical ATPase motif substitution that is necessary for unwinding needs to be tested to demonstrate that unwinding is indeed lost.

As discussed above, we have confirmed by several criteria that CMG can utilize ATPγS for unwinding, and has sequence dependency for being able to plow through to the end. This likely underlies the Costa lab 2018 CMG-forked DNA-ATPγS structure without using protein blocks. Hydrolysis by ATPγS is also consistent with CMG reaching the fork, or going beyond it, as we have shown CMG can’t move on ssDNA to reach the fork without hydrolysis (Wasserman et al., 2019; Georgescu et al., 2017).

Many enzymes that use ATP can also hydrolyze ATPγS (usually slowly), so ability of CMG to hydrolyze ATPγS is not surprising to us, but we have omitted this fact from this paper as it does not logically fit as a part of this study. Interestingly, Dan Herschlag has shown that a RNA helicase uses ATPγS reasonably efficiently, from which he implied that chemistry is not the rate limiting step (PMID 13130132). Also, the Benkovic lab found that Pol I uses dNMPaSPP nearly as well as dNTPs, and also used this observation to imply a slow conformation change is a rate limiting step in Pol I polymerization, and not the chemical step (PMID: 6351905).

Alternatively, the ATPγS data and claims could just be removed, as they are not central to the primary thrust of the paper.

We have taken the reviewer’s advice and omitted the ATPγS unwinding data.